# Biomimetic synergistic effect of redox site and Lewis acid for construction of efficient artificial enzyme

Haibin Si[1,4], Dexin Du[1,4], Chengcheng Jiao[1], Yan Sun[1], Lu Li[1,2] ✉ & Bo Tang [1,3] ✉

In enzymatic catalysis, the redox site and Lewis acid are the two main roles played by metal to assist amino acids. However, the reported enzyme mimics only focus on the redox-active metal as redox site, while the redox-inert metal as Lewis acid has, to the best of our knowledge, not been studied, presenting a bottleneck of enzyme mimics construction. Based on this, a series of highly efficient $M_xV_2O_5 \cdot nH_2O$ peroxidase mimics with vanadium as redox site and alkaline-earth metal ion ($M^{2+}$) as Lewis acid are reported. Experimental results and theoretical calculations indicate the peroxidase-mimicking activity of $M_xV_2O_5 \cdot nH_2O$ show a periodic change with the Lewis acidity (ion potential) of $M^{2+}$, revealing the mechanism of redox-inert $M^{2+}$ regulating electron transfer of V-O through non-covalent polarization and thus promoting $H_2O_2$ adsorbate dissociation. The biomimetic synergetic effect of redox site and Lewis acid is expected to provide an inspiration for design of enzyme mimics.

Enzyme is a kind of high-performance catalyst in biology, which controls most chemical reactions under physiological conditions[1,2]. It is important to simulate the structure and function of enzymes for the development of efficient enzyme-mimicking catalysts. In terms of structure, amino acids are the main body of enzymes, leading enzymes to catalyze reactions[3]. But in many cases, amino acids alone are not enough to handle all enzymatic reactions. In these cases, non-amino acid components can bind to the protein, thereby improving the reaction dynamics. Metal ions are present in nearly a third of all enzymes as cofactors, helping to catalyze biochemical reactions and perform specific physiological functions[4]. The role of metal ions in the enzymatic catalytic mechanism can be roughly divided into two groups, depending on whether the metal acts as a redox site in the catalytic process[5]. When the metal (redox-active metal ion) functions as a redox site, it is usually directly involved in one or more steps of an enzymatic reaction by obtaining/providing electrons from/to other reactive substances. Such as copper in superoxide dismutase[6,7], iron in peroxidase[8,9]. When the metal (redox-inert metal ion) functions as a non-redox site, it acts as a Lewis acid to reduce the activation energy by

activating the reaction substrate and stabilizing the intermediate or transition state. Such as magnesium in 6-hydroxymethyl-7,8-dihydropterin pyrophosphokinase, calcium in phosphoinositide phospholipase C[10,11].

The heme peroxidase superfamily, isolated from many plants, fungi, and bacteria, is a group of enzymes that use hydrogen peroxide to oxidize the reducing substrate[12,13] Horseradish peroxidase isoenzyme C (HRP-C), as the most studied isoenzyme of the peroxidase superfamily, its structure and function are typical[14,15]. It is generally acknowledged that the peroxidase contains high-spin $Fe^{3+}$, located in protoporphyrin IX in coordination with the proximal histidine ligand. As the redox center of peroxidase, $Fe^{3+}$ functions primarily by transferring electrons during enzymatic catalysis. At the same time, there is also redox-inert $Ca^{2+}$ in HRP-C with a content of 2 mol of $Ca^{2+}$/mol of enzyme. The $Ca^{2+}$ are on both sides of the protoporphyrin plane, which do not directly participate in electron transfer, but can significantly affect the catalytic activity of enzyme (left part of Fig. 1a). Haschke R H, Friedhoff J M. et al. found that after the removal of $Ca^{2+}$ in HRP-C, more than half of the activity of HRP-C could be lost[16,17]. Further study by Isao

[1]College of Chemistry, Chemical Engineering and Materials Science, Collaborative Innovation Center of Functionalized Probes for Chemical Imaging, Key Laboratory of Molecular and Nano Probes, Ministry of Education, Shandong Normal University, Jinan 250014, P. R. China. [2]Jinan Institute of Quantum Technology, Jinan 250101, P. R. China. [3]Laoshan Laboratory, Qingdao 266237, P. R. China. [4]These authors contributed equally: Haibin Si, Dexin Du. ✉e-mail: lilu5252@163.com; tangb@sdnu.edu.cn

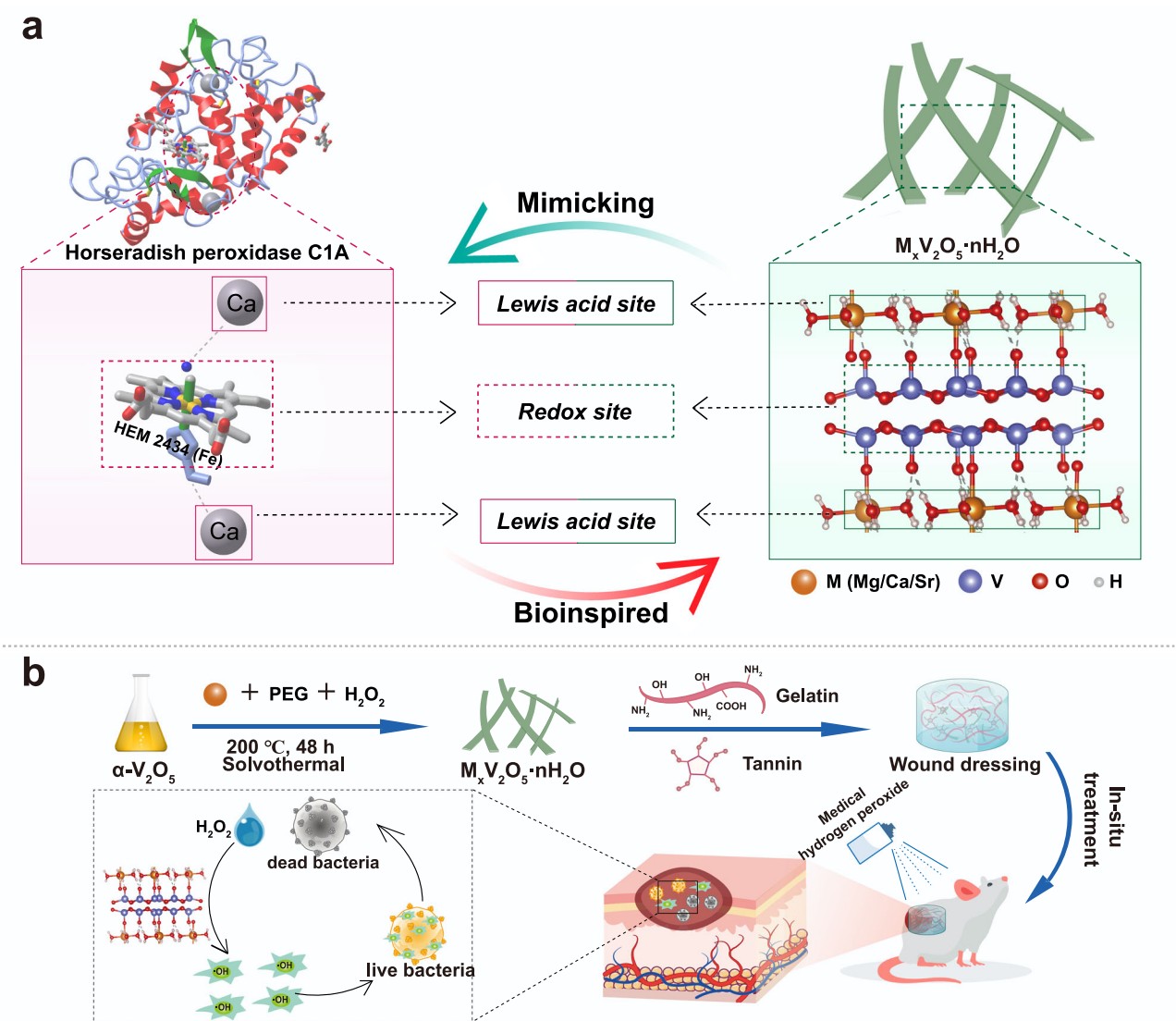

**Fig. 1 | Design peroxidase mimics with synergetic effect of redox-active vanadium and Lewis acidic alkali earth metal. a** Schematic illustration of the active site of natural horseradish peroxidase C1A (HRP C1A), in which $Fe^{3+}$ is located in protoporphyrin IX as redox site in coordination with the proximal histidine ligand and 2 mol $Ca^{2+}$ as Lewis acid site at the proximal and distal sites of redox site. The alkaline-earth metal M (M = Mg, Ca, Sr) and the active center V are connected by non-covalent bonds in bioinspired $M_xV_2O_5 \cdot nH_2O$, which resemble the Lewis acid site and the redox site in HRP C1A, respectively. **b** Bioinspired $M_xV_2O_5 \cdot nH_2O$ serves as the peroxidase mimics against bacteria for accelerating infected wound healing.

Morishima et al. showed that the removal of $Ca^{2+}$ significantly change the redox properties of HRP, and the redox potential of Fe(III)/Fe(II) experienced a cathodic shift of 30 mV. This may be due to the loss of $Ca^{2+}$ affects the structure of the imidazolyl ligand of the redox iron[13,18]. The similar results also be verified in wheat and barley peroxidase[19–21]. In addition, cationic peroxides extracted from the Mediterranean shrub Euphorbia characias have one strongly bound endogenous $Ca^{2+}$ [22]. Removal of $Ca^{2+}$ resulted in changes in the electronic structure of heme iron, and the activity of the Ca-free enzyme was about 2% of that of the natural enzyme. Moreover, the second $Ca^{2+}$ added make the enzyme efficiency parameter $k_{cat}/K_m$ was nearly 1000 times higher than that of native enzyme. These studies indicate that redox-inert metal ions play a key role in regulating the conformation and electronic structure of peroxidase.

Due to the important role of the metal ion in natural enzyme, many different types of metal-based nanomaterials, such as transition metal oxide nanomaterials[23,24], noble metal nanomaterials[25,26], metal organic frameworks (MOFs)[27], and single-atom catalysts[28–30], had been designed and synthesized for enzyme-mimicking catalysis. The metals in these enzyme mimics are usually transition elements with different valence states. These redox-active metal ions can achieve electron transfer in the catalytic process, which are a good simulation of the redox center in natural enzyme[28]. However, the enzyme mimics using redox-inert metal ions as Lewis acid inspired from natural enzymes have not been formally proposed and systematically explored. Considering the key role of redox-inert metal ions in adjusting the geometric and electronic structure of enzymes, the synergism of redox-active metal ions and redox-inert metal ions is expected to provide an inspiration for the design and synthesis of highly efficient enzyme-mimicking materials.

Inspired by the structure of natural peroxidase, the bionic synergetic structure with redox site and Lewis acid should have the following three characteristics: redox-active metal with high electron transfer ability as the main body of the enzyme-mimicking nanomaterials; redox-inert metal as Lewis acid site; the redox-inert metals exist in the main body of redox-active metal by weak bonding and have an effect on the geometric and electronic structure of the main body. Vanadium oxides such as $V_2O_5$[31,32], $VO_2$[33], $V_6O_{13}$[34] etc., have been

reported to have good peroxidase-mimicking activities, which mainly based on the multi-electron transfer ability of element V as redox center. In addition, vanadium oxides are a class of layered materials with highly anisotropic bonding, which are strongly bonded via covalent interaction within layers and stacked by weak van der Waals interactions between layers. The interlayer space allows for the intercalation of different guest species, such as cations, anions, organic molecules, etc., thus achieving structural and functional regulation[35–37]. Therefore, the layered vanadium oxide is an ideal material for constructing the bionic synergetic structure.

In this paper, with vanadium oxides ($V_2O_5$) as host material, introducing redox-inert alkaline-earth metal ion (M = Mg, Ca, Sr) guest between layers through intercalation, we constructed a series of $M_xV_2O_5\cdot nH_2O$ peroxidase mimics with a biomimetic synergistic structure of redox-active metal (redox site) and redox-inert metal (Lewis acid) (right part of Fig. 1a). The characterization show that $M_xV_2O_5\cdot nH_2O$ have a unique layered crystal structure with large interlayer spacing, a mixed oxidation state with high $V^{4+}/(V^{4+} + V^{5+})$ ratio, and abundant oxygen vacancies ($O_V$). $M_xV_2O_5\cdot nH_2O$ exhibited more significant peroxidase-mimicking activity compared with other typical peroxidase mimics, even some noble metal and single atom materials. Structural analysis, experimental results and theoretical calculations demonstrate that the excellent enzyme-mimicking activity of $M_xV_2O_5\cdot nH_2O$ can be ascribed to the increase of $O_V$ concentration caused by the change of coordination structure and the enhancement of $H_2O_2^*$ dissociation ability caused by the change of electronic structure after redox-inert $M^{2+}$ intercalation. Among them, this paper found that the enzyme-mimicking activity of $M_xV_2O_5\cdot nH_2O$ is based on the periodic change of Lewis acidity (ion potential) of intercalated $M^{2+}$, revealing the mechanism of the redox-inert $M^{2+}$ regulating electron transfer of V-O through non-covalent polarization and thus promoting OH* formation. Furthermore, $M_xV_2O_5\cdot nH_2O$ nanobelts were used as antibacterial additives in hydrogel as wound dressings due to their stable banded structure and high hydroxyl radical generation capacity. In vivo and in vitro experimental data show the wounding dressings containing $M_xV_2O_5\cdot nH_2O$ nanobelts possess the ability of good broad-spectrum antibacterial and promoting wound healing.

## Results and discussion

### Synthesis and characterization

$M_xV_2O_5\cdot nH_2O$ (M = Mg, Ca, Sr) was prepared by a simple reducing hydrothermal method[38]. Using commercially available $V_2O_5$ powder as precursor, then $H_2O_2$ was used as a reducing agent to reconstruct the crystal structure of $V_2O_5$ for the intercalation of different alkaline-earth metal ions by reacting with $MgCl_2\cdot6H_2O$, $CaCl_2$, $SrCl_2\cdot6H_2O$, respectively. In addition, the $V_2O_5$ nanobelt without intercalation as control sample was also as control samples prepared by same methods without adding alkaline-earth metal ions to investigate the effect of $M^{2+}$ intercalation on the structure and properties of $V_2O_5$. Scanning transmission microscope (SEM) and transmission electron microscopy (TEM) images showed that $V_2O_5$ powder were irregular particle with 1–3 μm (Supplementary Fig. 1A, 1 C). The $V_2O_5$ nanobelt exhibits unique 1D structure with length of 1 to 5 μm and width of 10 to 100 nm (Supplementary Fig. 1B, D). The lengths of $M_xV_2O_5\cdot nH_2O$ are between 1 to 10 μm and the widths are between 200 to 450 nm (Fig. 2A, B, Supplementary Fig. 2, 3). In addition, atomic force microscope (AFM) images show that the thicknesses of most $M_xV_2O_5\cdot nH_2O$ materials are in the range of 20–60 nm (Supplementary Fig. 4). Due to their large aspect ratio and nanoscale thickness, the $M_xV_2O_5\cdot nH_2O$ materials are also defined as $M_xV_2O_5\cdot nH_2O$ nanobelts. By high resolution transmission electron microscopy (HRTEM) analysis, the identical lattice spacing of $Mg_xV_2O_5\cdot nH_2O$, $Ca_xV_2O_5\cdot nH_2O$ and $Sr_xV_2O_5\cdot nH_2O$ nanobelts is 0.35 nm, corresponding to (100) lattice (Fig. 2C). It is worth noting that lattice disorder and mismatching can be observed in the HRTEM images of three kinds of $M_xV_2O_5\cdot nH_2O$ (Fig. 2D), which may come from

the surface oxygen vacancies ($O_V$) introduced by $M^{2+}$ intercalation. The concentration and formation of vacancies will be discussed in the subsequent experiments and characterization. The insets in Fig. 2C shows the selected area electron diffraction (SAED) images of $M_xV_2O_5\cdot nH_2O$. The uniform and well-ordered space lattices suggest the single-crystal diffraction pattern. Energy-dispersive X-ray spectroscopy (EDX) showed the existence and uniform distribution of M, V, and O in $M_xV_2O_5\cdot nH_2O$, and no element segregation was found when compared with HRTEM images (Fig. 2E). The chemical composition of $M_xV_2O_5\cdot nH_2O$ nanobelts were also investigated by inductively coupled plasma mass spectrometry (ICP-MS) and thermogravimetric analysis (TGA), and the results are shown in Supplementary Table 1 and Supplementary Fig. 5. The molecular formulas of the three kinds of $M_xV_2O_5\cdot nH_2O$ are $Mg_{0.22}V_2O_5\cdot0.75H_2O$ (MgVO), $Ca_{0.30}V_2O_5\cdot0.83H_2O$ (CaVO), $Sr_{0.34}V_2O_5\cdot0.72H_2O$ (SrVO), respectively.

The crystalline phases of $V_2O_5$ powder, $V_2O_5$ nanobelt, and $M_xV_2O_5\cdot nH_2O$ nanobelts were characterized by X-ray diffraction (XRD) to further verify the preparation of $V_2O_5$ nanobelt and $M_xV_2O_5\cdot nH_2O$ nanobelts. As shown in Fig. 2F, the XRD pattern of $V_2O_5$ powder matches the standard orthorhombic $V_2O_5$ (α-$V_2O_5$, JCPDS no. 41-1426, space group Pmmn). α-$V_2O_5$ has a typical two-dimensional (2D) layered structure, which facilitates metal ion insertion between layers (Fig. 2G). $V_2O_5$ nanobelt has similar crystal structure with $V_2O_5$ powder without crystal phase change. The XRD pattern of CaVO is well matched with the monoclinic phase $Ca_{0.24}V_2O_5\cdot H_2O$ (JCPDS no. 01-088-0579), which is a kind of layered material containing $Ca^{2+}$ and coordination water. Its cell parameters are as follows: a = 11.68 Å, b = 3.65 Å, c = 10.999 Å, β = 105.41°. No standard cards matching the XRD patterns of MgVO and SrVO were found, but the diffraction peaks corresponding to (001), (003), (004) and (005) crystal planes in the XRD patterns of the three kinds of $M_xV_2O_5\cdot nH_2O$ were similar, indicating that they have similar crystal structures. In particular, compared with $V_2O_5$ powder and $V_2O_5$ nanobelt, sharp diffraction peaks before 2θ = 10° appeared in $M_xV_2O_5\cdot nH_2O$ (Fig. 2F, Supplementary Fig. 6). The diffraction peaks near 2θ = 10° are generally attributed to the (001) crystal plane, which indicate that the crystal displays lamellar ordering as dominated by the pronounced (001) reflections[39]. With the intercalation of $M^{2+}$ and $H_2O$ in $M_xV_2O_5\cdot nH_2O$, the lattice spacing of (001) increases, causing the diffraction peak of (001) shift to a smaller angle[40]. By substituting the above diffraction peaks into Bragg equation $2d\sin\theta = n\lambda$, the crystal plane spacing of MgVO, CaVO and SrVO can be calculated as 10.54 Å, 10.66 Å, 10.23 Å, respectively. Compared with $V_2O_5$ (4.37 Å)[41], the larger crystal plane spacing further indicated the successful insertion of $Mg^{2+}$, $Ca^{2+}$, $Sr^{2+}$. The corresponding structure diagrams according to the XRD patterns and related crystal parameters are drawn in Fig. 2G. It can be seen from the schematic diagram that the α-$V_2O_5$ layer is composed of $VO_5$ square pyramids. However, square pyramids are connected up and down by sharing apexes point. Thus, if oxygen from adjacent square pyramids is included, the vanadium coordination geometry can be described as a distorted octahedra belonging to the saturated coordination[42]. $M_xV_2O_5\cdot nH_2O$ crystallized in the monoclinic space group $C2/m$. When α-$V_2O_5$ is converted to monoclinic $M_xV_2O_5\cdot nH_2O$, the V-O-V bilayers are separated into two adjacent V-O-V monolayers due to the elongated V-O distances. It has been shown that this is caused by the pre-intercalation of hydrated metal ions and free water molecular chains. Each V-O-V monolayer in $M_xV_2O_5\cdot nH_2O$ is composed of $\{V_aO_5\}$ tetragonal pyramids and $\{V_bO_6\}$ octahedrons, extending infinitely in the plane (Fig. 2G). In the interlayer of V-O-V bilayers, $M^{2+}$ coordinates with four water molecules to form planar square geometries, sharing edges with each other in a 1D chain. It can be seen that some V centers and M centers in $M_xV_2O_5\cdot nH_2O$ show unsaturated coordination, indicating that there may be abundant oxygen vacancies in $M_xV_2O_5\cdot nH_2O$[39,43]. $V_2O_5$ nanobelt has no obvious crystal structure change compared with $V_2O_5$ powder. This suggests that metal intercalation may be able to effectively induce crystal

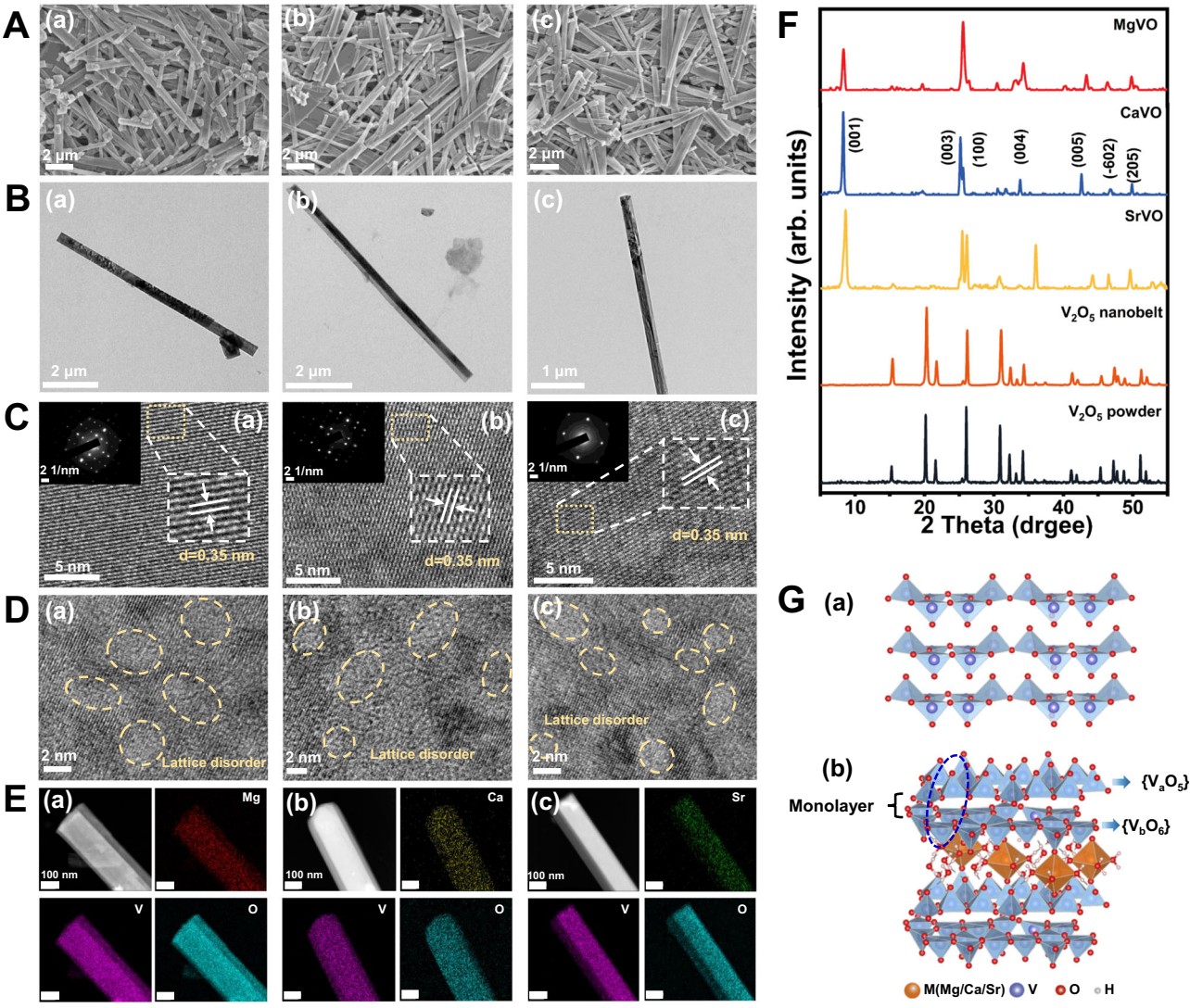

**Fig. 2 | Synthesis and characterization of $M_xV_2O_5 \cdot nH_2O$ nanobelts. A** SEM images, **B** TEM images, **C** HRTEM images (the inserts are SAED images) with clear lattice fringes and lattice disorder **D**, EDX-mapping (**E**) of MgVO (a), CaVO (b) and SrVO (c). **F** XRD patterns of MgVO, CaVO, SrVO, $V_2O_5$ nanobelt and $V_2O_5$ powder. **G** Crystal structure of α-$V_2O_5$ (a) and $M_xV_2O_5 \cdot nH_2O$ (b). Representative images are shown from three independent experiments with similar results (**A-E**).

structure change and oxygen vacancies generation, which is also consistent with related HRTEM characterization. It has been reported that transition metal center with unsaturated coordination have high catalytic activity, and abundant oxygen vacancies can provide adsorption sites for catalytic reactions[44,45], indicating that metal ion intercalation may bring additional catalytic activity to $M_xV_2O_5 \cdot nH_2O$.

To further understand the elemental composition and vacancies information of the prepared materials, X-ray photoelectron spectroscopy (XPS) analysis (Supplementary Fig. 7) was performed on $V_2O_5$ powder, $V_2O_5$ nanobelt and $M_xV_2O_5 \cdot nH_2O$ nanobelts. Compared with $V_2O_5$ powder and $V_2O_5$ nanobelt, $M_xV_2O_5 \cdot nH_2O$ nanobelts show new peaks at 1304.2 eV, 345.7 eV, and 351.0 eV, which corresponding to Mg $1s$, Ca $2p_{3/2}$, Ca $2p_{1/2}$ and Sr $3d_{3/2}/3d_{5/2}$, respectively. This further confirmed the successful intercalation of $M^{2+}$ in $M_xV_2O_5 \cdot nH_2O$ nanobelts. The five materials all show two similar characteristic peaks between 512–528 eV, corresponding to the spin orbit peaks of V $2p_{3/2}$ and V $2p_{1/2}$ respectively. Compared with $V_2O_5$ powder, the V $2p_{3/2}$ and V $2p_{1/2}$ of $M_xV_2O_5 \cdot nH_2O$ nanobelts show prominent asymmetry, indicating that V exists in a larger $V^{4+}/(V^{4+} + V^{5+})$ ratio in $M_xV_2O_5 \cdot nH_2O$ nanobelts (Fig. 3A). The deconvolution of the peaks show that the proportion of $V^{4+}/(V^{4+} + V^{5+})$ in $M_xV_2O_5 \cdot nH_2O$ nanobelts reach 35.7% 34.6% and 31.5%, respectively, significantly higher than 18.6% in $V_2O_5$ powder, indicating

that more $V^{5+}$ are reduced to $V^{4+}$ during the generation of $M_xV_2O_5 \cdot nH_2O$ nanobelts (Fig. 3C). However, the $V^{4+}/(V^{4+} + V^{5+})$ ratio in $V_2O_5$ nanobelt with the same dose of reducing agent added in the preparation process is only 21.3%, meaning that the insertion of $M^{2+}$ is conducive to the existence of high $V^{4+}/(V^{4+} + V^{5+})$ ratio. The relative content of oxygen vacancies in $V_2O_5$ nanobelt and $M_xV_2O_5 \cdot nH_2O$ nanobelts were further analyzed based on O $1s$ high resolution spectra. As shown in Fig. 3B, the O $1s$ spectrum can be fitted to three peaks of 529.5 eV, 530.5 eV, and 532 eV, which belong to the oxygen vacancies and surface-adsorbed oxygen ($O_S$) and lattice oxygen ($O_L$), respectively. As expected, the relative content of $O_V$ in $M_xV_2O_5 \cdot nH_2O$ nanobelts reach 22.76%, 17.96%, and 17.65% with the increase of $V^{4+}/(V^{4+} + V^{5+})$ ratio respectively, which higher than $V_2O_5$ powder (5.34%) and $V_2O_5$ nanobelt (11.2%) (Fig. 3C). The reconstruction of the structure and the generation of $O_V$ are further confirmed by the use of electron spin resonance (ESR) spectra. As shown in Fig. 3D, $M_xV_2O_5 \cdot nH_2O$ nanobelts transmit an intense resonance signal at g = 1.97, corresponding to the captured electrons in $O_V$, whereas only a weak signal is detected in $V_2O_5$ powder and $V_2O_5$ nanobelt. The results of ESR are consistent with those of XPS, and verified with HRTEM and XRD. In order to further understand the bonding structure of the prepared $M_xV_2O_5 \cdot nH_2O$ nanobelts, Fourier transform infrared (FT-IR) spectroscopy and Raman

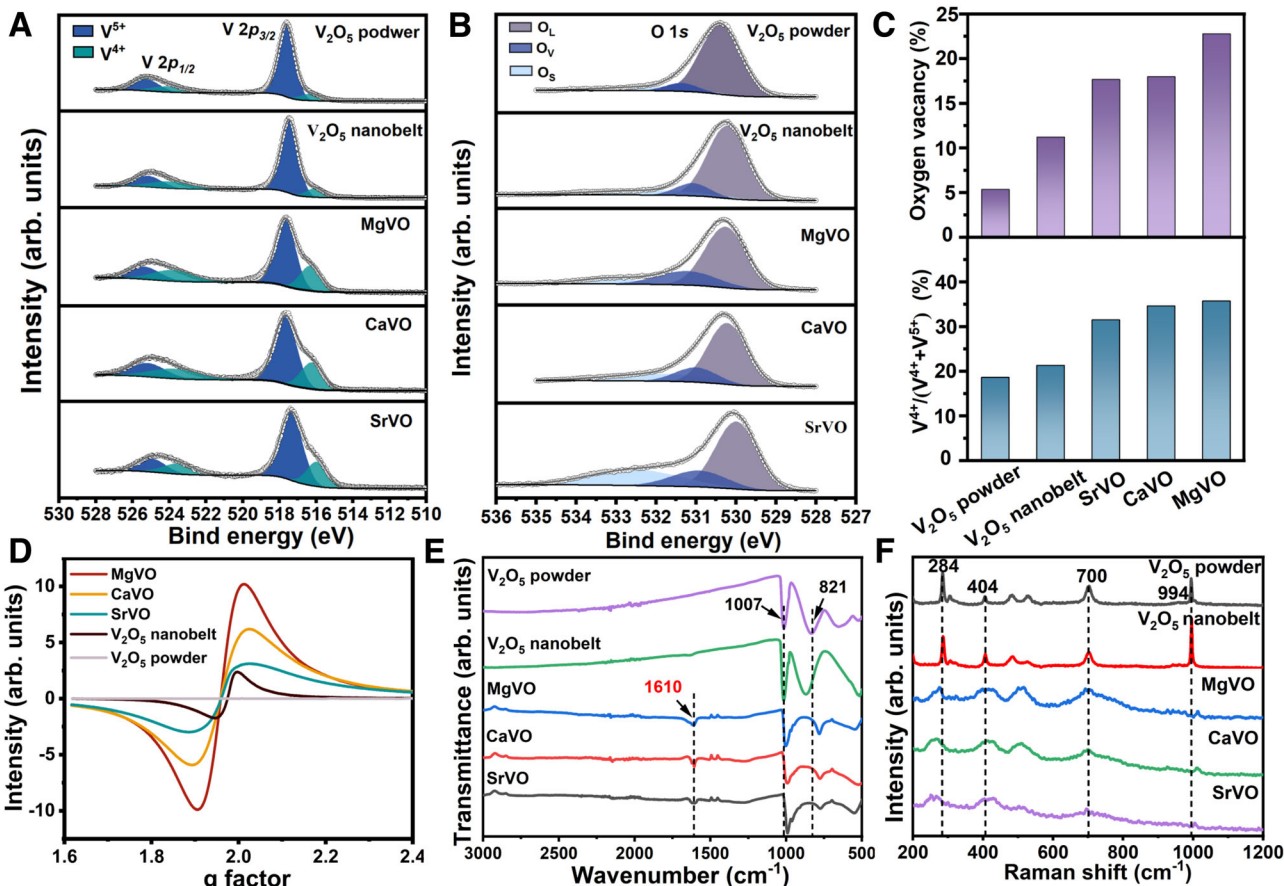

**Fig. 3 | Characterization of $M_xV_2O_5 \cdot nH_2O$ nanobelts. A** XPS spectra of V $2p$ and **B** O $1s$ of, $V_2O_5$ powder, $V_2O_5$ nanobelt, MgVO, CaVO, and SrVO. **C** Quantitative analysis of the $V^{4+}/(V^{4+} + V^{5+})$ ratio and relative content of $O_V$ of $V_2O_5$ powder, $V_2O_5$ nanobelt, MgVO, CaVO, and SrVO. **D** The analysis of $O_V$ of $V_2O_5$ powder, $V_2O_5$ nanobelt, MgVO, CaVO, and SrVO by ESR spectroscopy. **E** FTIR spectra (1007 cm$^{-1}$: symmetric stretching vibration of V = O; 821 cm$^{-1}$: stretching vibration of V-O-V; 1610 cm$^{-1}$: vibration of O-H) and (**F**) Raman spectra of $V_2O_5$ powder, $V_2O_5$ nanobelt, MgVO, CaVO, and SrVO. (994 cm$^{-1}$: symmetric stretching vibration of V = O; 700 and 404 cm$^{-1}$: stretching vibration of V-O; 284 cm$^{-1}$: vibration of V = O).

spectroscopy were used to analyze them. The attributions of major characteristic peaks are shown in Fig. 3E, F, which is consistent with relevant reports[46].

## Highly specific and efficient enzyme-mimicking activity of $M_xV_2O_5 \cdot nH_2O$ nanobelts

The $M_xV_2O_5 \cdot nH_2O$ nanobelts prepared in this paper are expected to present excellent catalytic performance due to the unsaturated coordination of vanadium and abundant oxygen vacancies caused by $M^{2+}$ intercalation. The peroxidase-mimicking activity of $M_xV_2O_5$ nanobelts were studied by using 3, 3', 5, 5'-tetramethylbenzidine (TMB), 2, 2'-azino-bis (3-ethylbenzothiazoline-6-sulfonic acid) (ABTS) and o-Phenylenediamine (OPD) as substrates for typical color reaction in the presence of $H_2O_2$. As shown in Fig. 4A, $M_xV_2O_5 \cdot nH_2O$ nanobelts can effectively catalyze the oxidation of TMB in the presence of $H_2O_2$, showing obvious peroxidase-mimicking activity. Experiments also show that $M_xV_2O_5 \cdot nH_2O$ nanobelts exhibit optimal peroxidase-mimicking catalytic activity under weak acidic environment (pH=5.0) and near room temperature (10–80 °C) (Supplementary Fig. 8). The $M_xV_2O_5 \cdot nH_2O$ nanobelts maintain stable crystal structures and morphologies within 15 days at pH=5.0 (Supplementary Fig. 9, 10). Correspondingly, $V_2O_5$ powder show no detectable peroxidase-mimicking activity, while $V_2O_5$ nanobelt without $M^{2+}$ intercalation show significantly lower catalytic activity than $M_xV_2O_5 \cdot nH_2O$ nanobelts under the same conditions. These results indicate that the activated peroxidase-mimicking activity of $M_xV_2O_5 \cdot nH_2O$ nanobelts may be related to the composition and structure changes after $M^{2+}$

intercalation. The above results were also confirmed when ABTS and OPD were used as substrates (Supplementary Fig. 11). Meanwhile, cyclic voltammetry was used to verify the catalytic activity of $M_xV_2O_5 \cdot nH_2O$ on the substrate[26,47]. As shown in Supplementary Fig. 12, 13, when $H_2O_2$ and TMB are added to the electrolyte, the currents corresponding to the redox of V with different valence states are significantly reduced, showing obvious catalytic responses. In the absence of $H_2O_2$, $M_xV_2O_5 \cdot nH_2O$ nanobelts can not effectively catalyze the oxidation of TMB/ABTS/OPD, indicating that it has almost no oxidase-mimicking activity (Supplementary Fig. 14). In addition, the dissolved oxygen test and the nitrogen blue tetrazole (NBT) colorimetry showed that $M_xV_2O_5 \cdot nH_2O$ nanobelts show no obvious catalase-mimicking activity and superoxide dismutase-mimicking activity (Supplementary Fig. 14). In addition, previous studies have shown that some vanadium-based nanozymes also demonstrate haloperoxidase and glutathione peroxidase activities[48,49]. Therefore, we detected the haloperoxidase-mimicking activity and glutathione peroxidase-mimicking activity of $M_xV_2O_5 \cdot nH_2O$. As shown in Supplementary Fig. 15 and 16, $M_xV_2O_5 \cdot nH_2O$ show glutathione-mimicking peroxidase activity, and no significant haloperoxidase-mimicking activity was detected.

According to the peroxidase catalytic mechanism, the ability of $M_xV_2O_5 \cdot nH_2O$ to catalyze the decomposition of $H_2O_2$ to produce ·OH was verified by ESR spectra. Due to the short life and high chemical activity of ·OH, 5, 5-dimethyl-1-pyrrolin-n-oxide (DMPO) was used as ·OH radical catcher to evaluate the production of ·OH. As shown in Fig. 4B, in the presence of $H_2O_2$, the ESR spectra of $M_xV_2O_5 \cdot nH_2O$

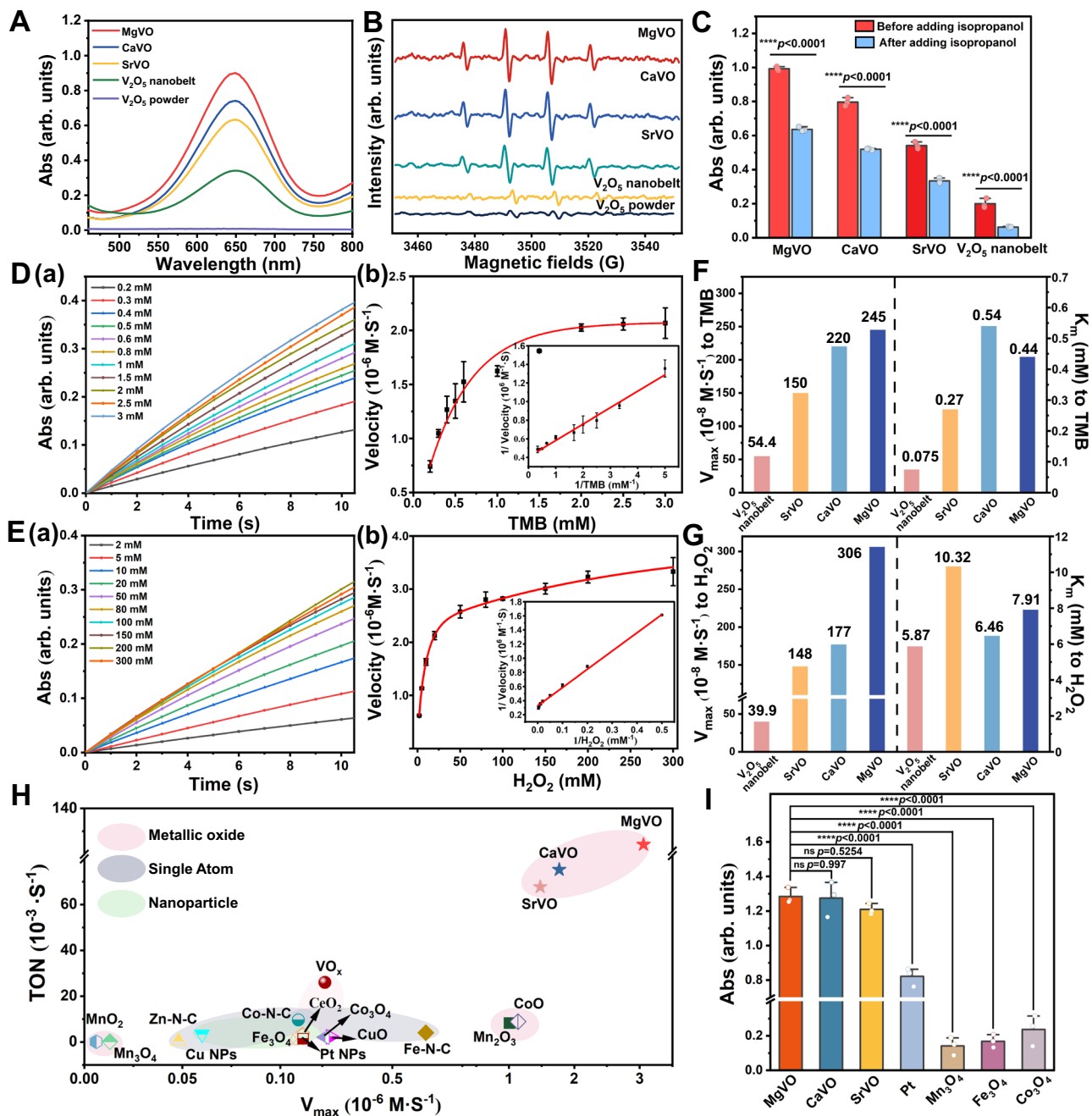

**Fig. 4 | Detection of enzyme-mimicking activity and steady-state kinetics of $M_xV_2O_5 \cdot nH_2O$. A** UV-vis absorption spectra of oxTMB in the presence of $H_2O_2$ with $M_xV_2O_5 \cdot nH_2O$, $V_2O_5$ powder, $V_2O_5$ nanobelt, respectively. **B** The generation of $\cdot OH$ testified by ESR spectra of $M_xV_2O_5 \cdot nH_2O$, $V_2O_5$ powder, $V_2O_5$ nanobelt in the presence of $H_2O_2$. **C** Absorbance of oxTMB in the presence of $H_2O_2$ and $M_xV_2O_5 \cdot nH_2O/$ $V_2O_5$ nanobelt after adding isopropanol. $n = 3$ experimental replicates. Data are presented as mean values ± SD. The data were analyzed by using a one-sided unpaired $t$-test. ****$P < 0.0001$. **D** Time-dependent absorbance of oxTMB (a), reaction rate (b) with different concentrations of TMB and fixed concentrations of MgVO/$H_2O_2$, and the corresponding double reciprocal (Lineweaver-Burk) plots (inset). **E** Time-dependent absorbance of oxTMB (a), reaction rate (b) with different

concentrations of $H_2O_2$ and fixed concentrations of MgVO/TMB, and the corresponding double reciprocal (Lineweaver-Burk) plots (inset). $n = 3$ experimental replicates (**D**, **E** (b)). Data are presented as mean values ± SD (**D**, **E** (b)). $V_{max}$ and $K_m$ values of $M_xV_2O_5 \cdot nH_2O/V_2O_5$ nanobelt towards to TMB (**F**) and (**G**) $H_2O_2$. **H** The comparison of the TON and $V_{max}$ values of $M_xV_2O_5 \cdot nH_2O$ with other recently reported catalysts. **I** The comparison of the relative catalytic activity of $M_xV_2O_5 \cdot nH_2O$ and synthesized peroxidase mimics. $n = 3$ experimental replicates. Data are presented as mean values ± SD. Data were analyzed by one-way ANOVA with Turkey's multiple comparisons test, ns represented no statistical difference, ****$P < 0.0001$.

show an obvious characteristic spectroscopy of 1:2:2:1 intensity for DMPO/$\cdot OH$ adducts, which proved the generation of $\cdot OH$ in the $M_xV_2O_5 \cdot nH_2O/H_2O_2$ system. To further verify the generation of $\cdot OH$, isopropyl alcohol as $\cdot OH$ scavenger was added to the $M_xV_2O_5 \cdot nH_2O/$ $H_2O_2/TMB$ catalytic reaction system, the absorbance of oxTMB in the reaction system decreased significantly (Fig. 4C), illustrating $\cdot OH$

is the main reactive species in the enzymatic reaction system of $M_xV_2O_5 \cdot nH_2O$.

In order to evaluate the peroxidase-mimicking activity of $M_xV_2O_5 \cdot nH_2O$ nanobelts, the steady-state kinetic analysis of $M_xV_2O_5 \cdot nH_2O$ was carried out in proper concentration range of TMB and $H_2O_2$. Figure 4D, E and Supplementary Fig. 17–19 show the

absorption spectra of catalytic oxidation of TMB by $M_xV_2O_5 \cdot nH_2O$ at different concentrations of TMB or $H_2O_2$. The XPS analysis of $M_xV_2O_5 \cdot nH_2O$ was performed before and after the reaction. It can be seen from Supplementary Fig. 20 that $M_xV_2O_5 \cdot nH_2O$ remain relatively stable under catalytic conditions. According to the initial reaction rate calculated in Fig. 4D (a), 4E (a), typical Michaelis-Menten curves of TMB and $H_2O_2$ can be obtained respectively (Fig. 4D (b), 4E(b)). Next, the reciprocal of substrate concentration and initial reaction rate could be used to prepare the double reciprocal plot (Lineweaver-Burk) to obtain the maximal reaction velocity ($V_{max}$) and Michaelis−Menten constant ($K_m$). Among them, $V_{max}$ is an important parameter to evaluate the catalytic rate of enzymatic reaction, and $K_m$ is an important indicator of enzyme affinity to substrates. As shown in Fig. 4F, when TMB was used as substrate, $V_{max}$ of MgVO, CaVO and SrVO nanobelts are 245, 220, 150 ($10^{-8}$ M·s$^{-1}$), $K_m$ are 0.44, 0.54, 0.27 mM, respectively; When $H_2O_2$ was used as substrate (Fig. 4G), the $V_{max}$ of MgVO, CaVO and SrVO nanobelts are 306, 177, 148 ($10^{-8}$ M·s$^{-1}$), and $K_m$ are 7.91, 6.46, 10.32 mM, respectively. Comparing with natural horseradish peroxidase (HRP) and classical $Fe_3O_4$ (Supplementary Table 2), the $K_m$ of $M_xV_2O_5 \cdot nH_2O$ for the two substrates are similar to that of HRP and $Fe_3O_4$, but the $V_{max}$ of $M_xV_2O_5 \cdot nH_2O$ are more than one order of magnitude higher than that of HRP and $Fe_3O_4$ under their respective optimal catalytic conditions. In order to systematically evaluate its catalytic performance, we compared the $V_{max}$ and TON values (the maximum number of conversing substrates via the mole concentration of metal in the whole nanomaterials[50]) of $M_xV_2O_5 \cdot nH_2O$ with currently reported classical peroxidase mimics, including Zn-N-C[51], Pt NPs[52], $CeO_2$[53], $Fe_3O_4$[53], and $MnO_2$[53], etc (Supplementary Table 3), the results indicate $M_xV_2O_5 \cdot nH_2O$ display the optimal catalytic performances in comparison with metal oxides-based nanozymes, even some noble metal nanozymes and single atom nanozymes (Fig. 4H). In order to eliminate data differences caused by different test conditions, four typical peroxidase-mimicking materials were synthesized by referring to the methods in the literature (Supplementary Fig. 21, 22), and catalytic oxidation of TMB was tested under the same test conditions. The results of Fig. 4I directly confirm that $M_xV_2O_5 \cdot nH_2O$ nanobelts presented high efficiency peroxidase-mimicking activity.

## The synergistic effect of redox V and redox-inert M in enzyme-mimicking catalysis

Theoretical calculations and some experimental studies have showed that the catalytic reaction paths of POD-like nanozymes can be classified as two types according to dissociation mode of $H_2O_2$ adsorbate ($H_2O_2^*$)[54,55]. The path 1 is that the dissociated $H_2O_2^*$ generates hydroxyl adsorbate (OH*) and hydroxyl radical (·OH), then generates $H_2O$ and oxidized substrate to complete the cycle[30,51,56–58]. Because it is similar to the well-known Fenton reaction, this mechanism is also known as the Fenton-like mechanism and is widely accepted to explain the POD-like activity of many materials. The path 2 usually does not involve the formation of ·OH. $H_2O_2^*$ first generates two OH*, which can directly oxidize the substrate under acidic conditions[59,60]. Alternatively, OH* can also be converted to O* and $H_2O^*$ through a hydrogen transfer reaction, with O* oxidizing the substrate[53,57,61,62]. This path that the adsorbed ROS directly acting on substrates has attracted more and more attention because it is similar to the catalytic mechanism of natural peroxidase (ferryl oxo species). In view of the direct detection of ·OH produced under catalysis by ESR spectroscopy and the scavenging verification ·OH by isopropyl alcohol, the possible catalytic mechanism of $M_xV_2O_5 \cdot nH_2O$ is proposed following path 1. Therefore, the reaction mechanism was studied from the free energy, charge density difference, and band structure through density functional theory calculation. Subsequently, combined with the changes in the structure and catalytic activity of $M_xV_2O_5 \cdot nH_2O$ after $M^{2+}$ intercalation, the synergistic effect of redox-inert $M^{2+}$ with redox-active V is further discussed.

Four crystal models including $V_2O_5$, MgVO, CaVO and SrVO were established to simulate $V_2O_5$ nanobelt and three kinds of $M_xV_2O_5 \cdot nH_2O$ nanobelts. In addition, to exclude the effect of crystal structure change ($V_2O_5$ and $M_xV_2O_5 \cdot nH_2O$) on the catalytic process, an additional $V_2O_5 \cdot nH_2O$ crystal model with only $H_2O$ intercalation was also established to study the effect of $M^{2+}$ in catalysis comparison with $M_xV_2O_5 \cdot nH_2O$. At the same time, since $V_2O_5 \cdot nH_2O$ have unsaturated coordination structure relative to $V_2O_5$ like $M_xV_2O_5 \cdot nH_2O$, the contrast between $V_2O_5 \cdot nH_2O$ and $V_2O_5$ can be used to separately analyze the role of oxygen vacancies caused by unsaturated coordination in catalysis. The top and side views of the above five crystal models are shown in Supplementary Fig. 23. The possible catalytic mechanism of $M_xV_2O_5 \cdot nH_2O$ catalyzing $H_2O_2$ decomposition is shown in Fig. 5A. Taking MgVO as an example, $H_2O_2$ tends to preferentially adsorb on the active site V of MgVO, and then the activated $H_2O_2$ is uniformly dissociated into two OH*. Among them, one OH* desorbs from the adsorption position to form ·OH, and the other OH* binds with a protonated hydrogen atom to form $H_2O$ molecule. MgVO returns to its initial state after $H_2O$ desorption. According to this mechanism, the free energy changes in the reaction process of the five models was shown in Fig. 5B.

Combined with the free energy profile and related literature[51,60], it can be inferred that there are two key steps in this catalytic path, $H_2O_2^*$ dissociation and OH* desorption. For $V_2O_5$ before intercalation, the energy of $H_2O_2^*$ dissociation is 2.03 eV, indicating that the process is endothermic. For $V_2O_5 \cdot nH_2O$ after $H_2O$ intercalation, the energy of $H_2O_2^*$ dissociation decreased to 1.09 eV, indicating that the dissociation is promoted. This may be owed to the oxygen vacancies caused by unsaturated coordination in $H_2O$-intercalated $V_2O_5 \cdot nH_2O$. In spite of this, the $H_2O_2^*$ dissociation for $V_2O_5 \cdot nH_2O$ is still an endothermic process, and the reaction is not favorable. After Mg/Ca/Sr intercalation, the $H_2O_2$ dissociations are obviously promoted, and their free energies all become negative, that is, the process changes from endothermic to exothermic (Fig. 5B, C). Since the energy of $H_2O_2^*$ dissociation is determined by the difference between the adsorption energy ($E_{ad}$) of OH* and $H_2O_2^*$, $E_{ad}$ (OH*) can be considered as descriptor of the process under the premise that $E_{ad}$ ($H_2O_2^*$) are similar. Therefore, compared with $V_2O_5$ and $V_2O_5 \cdot nH_2O$, $M^{2+}$ intercalation significantly enhanced the adsorption of OH*, thus promoting the dissociation of $H_2O_2^*$. Further comparing the effect of different intercalation ions, CaVO showed a stronger ability than MgVO and SrVO to promote $H_2O_2^*$ dissociation, which is inconsistent with the order of $V_{max}$ obtained by experiments. Considering that strong OH* adsorption is beneficial to the dissociation of $H_2O_2^*$, but not conducive to the desorption of OH*, the catalytic activities may also be affected by OH* desorption. According to the free energy profile, the energy of OH* desorption on MgVO is only 0.196 eV, meaning desorption is relatively simple. But the energies of CaVO and SrVO during desorption are 0.503 eV and 0.484 eV, respectively (Fig. 5B, D), which is not favorable for the dissociation of OH*. On the whole, considering the energy from $H_2O_2^*$ to OH* (Step II & III), the order of MgVO (0.165 eV) < CaVO (0.216 eV) < SrVO (0.471 eV) is presented (Fig. 5B, D). It is worth noting that the energy change (Step II & III) is positively correlated with the experimental $V_{max}$, and shows the consistency with the periodic change of M. Besides the above mechanism involving releasing ·OH, there are many reports on mechanism that the OH* oxidizes substrates directly under acidic conditions[59,60]. In this path, $H_2O_2^*$ first generates two OH*, and then the reductive substrates (such as TMB) continuously providing reductive hydrogen [H] to the two adsorbed OH*. Since the presence of the substrate, it is considered to be more similar in energy to the actual catalytic process. Based on this, we calculate the free energy profile under this path. As shown in Supplementary Fig. 24, the dissociation of $H_2O_2$ is not affected after the substrate is added. Compared with $V_2O_5$ and $V_2O_5 \cdot nH_2O$, the energy of $H_2O_2$ dissociation decreases significantly after Mg/Ca/Sr

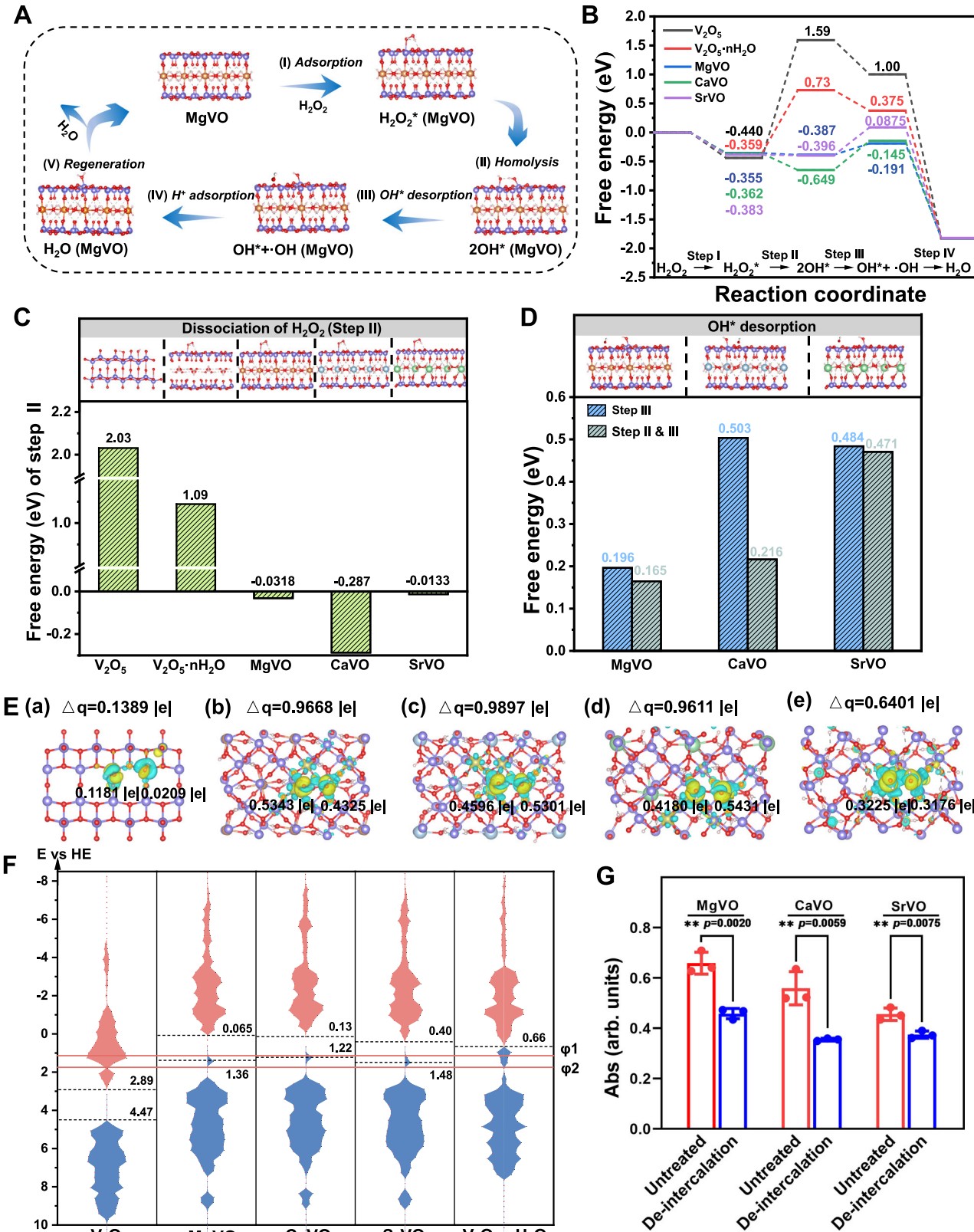

**Fig. 5 | DFT calculations and the proposed mechanism for the peroxidase-mimicking of $M_xV_2O_5 \cdot nH_2O$. A** The possible reaction pathway on $M_xV_2O_5 \cdot nH_2O$, taking MgVO for example. (Purple: V atom; Red: O atom; White: H atom; Yellow: Mg atom). **B** Free energy diagram of the proposed reaction pathways for $V_2O_5$, $V_2O_5 \cdot nH_2O$ and $M_xV_2O_5 \cdot nH_2O$ models. **C** Free energy change of dissociation of $H_2O_2$ (Step II) on different models. (Blue, Ca atom; Green: Sr atom). **D** Free energy change of OH* desorption (Step III or Step II & III) on $M_xV_2O_5 \cdot nH_2O$ models. **E** The differential charge analysis of (a) $V_2O_5$, (b) MgVO, (c) CaVO, (d) SrVO and (e) $V_2O_5 \cdot nH_2O$ with 2OH* models (cyan and yellow represent charge depletion and accumulation, respectively). **F** Calculated electronic density of states with the energies of FMOs marked. **G** Comparison of peroxidase-mimicking activity of $M_xV_2O_5 \cdot nH_2O$ before and after de-intercalation. $n$ = 3 experimental replicates. Data are presented as mean values ± SD. Data were analyzed by using a one-sided unpaired $t$-test, **$P < 0.01$.

intercalation. This is consistent with the result in path 1 (Fig. 5B). The reaction of OH* with TMB-H$^+$ becomes completely spontaneous due to the formation of H$_2$O. Therefore, from the perspective of free energy change, Ead (OH*), that is, the stability of OH* on the catalyst surface, is key to determine the catalytic reaction activity in both path 1 (Fig. 5B) and path 2 (Supplementary Fig. 24). The redox-inert M$^{2+}$ are not considered to be catalytic centers like redox-active V. Therefore, the synergistic function of redox-inert M$^{2+}$ through regulating the electronic structure of redox-active V may be the main way to improve their enzyme-like catalytic activities. The interlayer spacing of M$_x$V$_2$O$_5$·nH$_2$O indicated that the intercalated M$^{2+}$ and the host was connected by non-covalent bonds[43]. Therefore, the effect of intercalated M$^{2+}$ on the electronic structure of the crystal is mainly polarization according to their Lewis acidity, which can usually be evaluated by the ionic potential. The Ionic potential ($\Phi$) is defined as

$$\Phi = Z_{eff}/r = (Z - \sigma)/r \qquad (1)$$

where $Z_{eff}$ is the effective nuclear charge, $\sigma$ is the screening constant, and r is the ionic radius[63]. Among them, the effective nuclear charge is a periodic property and depends upon the electronic shell structures although it is only qualitatively conceived and measured semiempirically[64]. Since Mg$^{2+}$, Ca$^{2+}$ and Sr$^{2+}$ belong to the IIA in the periodic table of the elements, they have the same ionic charge and valence shell electron configuration, so there is little difference in their $Z_{eff}$. Therefore, the ionic potential of Mg$^{2+}$, Ca$^{2+}$, Sr$^{2+}$ is inversely proportional to their ionic radius. With the increase of ionic potential, the Sr$^{2+}$, Ca$^{2+}$ and Mg$^{2+}$ in the interlayer of M$_x$V$_2$O$_5$·nH$_2$O showed a periodic enhancement of polarization, thus polarizing the inherent V-O bond in the crystal and making V-OH* more stable.

To confirm the above theory, the differential charge density calculation was used to and analyze the electron transfer from M$_x$V$_2$O$_5$·nH$_2$O to OH* to determine the stability of OH*. As shown in Fig. 5E, there is basically no electron transfer (0.1181 and 0.0209 |e| respectively) between V and two OH* in V$_2$O$_5$ before intercalation, indicating that there is no stable interaction between V and OH*. After H$_2$O intercalation (V$_2$O$_5$·nH$_2$O), the electrons transferred between V and two OH* is significantly enhanced, reaching 0.3225 and 0.3176 |e| respectively. After M$^{2+}$ intercalation, the number of transferred electrons increased further (0.5343/0.4325 |e|; 0.4596/0.5301 |e|; 0.4180/0.5431 |e|), indicating the stable bonding interaction between V and OH*. On the whole, the order of charge transfer from the V to two OH* is CaVO (0.9897 |e|) > MgVO (0.9668 |e|) > SrVO (0.9611 |e|) > V$_2$O$_5$·nH$_2$O (0.6401 |e|) > V$_2$O$_5$ (0.1389 |e|), where the charge transfers after M$^{2+}$ intercalation are significantly greater than that of V$_2$O$_5$ and V$_2$O$_5$·nH$_2$O, which corresponds to the energy of H$_2$O$_2$* dissociation. Then, the influence of Lewis acid M$^{2+}$ on the intrinsic electronic structure of redox-active V is further analyzed by calculating the average bader charge of V before and after intercalation. As shown in Supplementary Fig. 25, the bader charge around V in V$_2$O$_5$ before intercalation is 2.1670 |e|. After H$_2$O intercalation, the bader charge around V is 2.1730 |e|, and no significant change occurs. After M$^{2+}$ intercalation, the average bader charge around V decreases to 2.1379, 2.1264 and 2.1242 |e|, respectively. Since M$^{2+}$ acts directly on the coordinated O of V through electrostatic attraction, the polarization of M$^{2+}$ on the V-O bond results in the decrease in electron density around V. In general, V with low electron density will be more inclined to produce stable V-OH*. Thus, the Lewis acid M$^{2+}$ promotes the H$_2$O$_2$* homolysis by adjusting the electronic structure of V.

In order to further understand the effect of Lewis acid intercalation on enzyme-mimicking catalytic activity, the band structure before and after intercalation were calculated and discussed combing with the POD-like catalytic redox potential[65,66]. The typical reaction catalyzed by peroxidase nanozyme is as follows: H$_2$O$_2$ + 2TMB + 2H$^+$ = 2H$_2$O + 2TMB$^+$, where TMB$^+$ is the oxidation state of TMB. The reaction can be divided

into two half reactions as follows:

$$TMB^+ + e^- = TMB, \varphi_1;$$
$$1/2H_2O_2 + H^+ + e^- = H_2O, \varphi_2$$

The reduction potential of TMB$^+$/TMB ($\varphi_1$) used is about 1.13 V, referring to a well-established value in the literature, and the standard reduction potential of H$_2$O$_2$/H$_2$O ($\varphi_2$) is 1.776 V[67,68]. As can be seen from Fig. 5F, the frontier molecular orbitals (FMO) of V$_2$O$_5$ before intercalation, including the valence band maximum (VBM) and the conduction band minimum (CBM), display more positive energy than $\varphi_2$, indicating that electrons can be transferred from TMB to V$_2$O$_5$, but cannot be transferred from V$_2$O$_5$ to H$_2$O$_2$ to complete the catalytic reaction. After Lewis acid M$^{2+}$ intercalation, the energies of the FMO of M$_x$V$_2$O$_5$·nH$_2$O move significantly toward the negative direction, indicating that the reduction abilities of M$_x$V$_2$O$_5$·nH$_2$O are significantly enhanced. Among them, the energies of VBM is close to or less negative than $\varphi_2$, indicating that electrons can be passed from TMB to V$_2$O$_5$, and then to H$_2$O$_2$ to complete the catalysis. The H$_2$O intercalated V$_2$O$_5$·nH$_2$O shows a more negative FMO energy, and electrons cannot transfer from TMB to V$_2$O$_5$.

According to the reported literatures, the redox-inert Ca$^{2+}$ as cofactors in natural peroxidase with Lewis acidity play a key role in the catalytic performance, which is mainly manifested in the loss of Ca$^{2+}$ leading to significant reduction of catalytic activity[16,17]. The structural similarities between M$_x$V$_2$O$_5$·nH$_2$O and natural peroxidase encouraged us to further investigate whether the redox-inert M$^{2+}$ could also act as cofactors like Ca$^{2+}$ of natural peroxidase to affect the peroxidase-mimicking activity of M$_x$V$_2$O$_5$·nH$_2$O. Thus, M$_x$V$_2$O$_5$·nH$_2$O were placed in heated alkaline liquid to remove the inserted M$^{2+}$ to achieve deintercalation. The results of ICP-MS showed that the molar ratios of Mg/V, Ca/V, and Sr/V in M$_x$V$_2$O$_5$·nH$_2$O decreased to 0.08/1, 0.13/1, and 0.14/1 respectively (Supplementary Fig. 26, Supplementary Table 4), confirming partial removal of M$^{2+}$ in M$_x$V$_2$O$_5$·nH$_2$O. The catalytic capacity of M$_x$V$_2$O$_5$·nH$_2$O for TMB oxidation before and after de-intercalation were then compared. From the result shown in Fig. 5G, the catalytic performance of M$_x$V$_2$O$_5$·nH$_2$O after de-intercalation show obvious decrement owing to the decreased concentration of M$^{2+}$. The SEM and XRD results show that the morphology and crystal structure of M$_x$V$_2$O$_5$·nH$_2$O after partial de-intercalation have no obvious changes (Supplementary Fig. 27, 28). The above results showed that the redox-inert M$^{2+}$ in M$_x$V$_2$O$_5$·nH$_2$O play similar roles to the Ca$^{2+}$ in natural peroxidase in regulating the catalytic reaction. Benefiting from these, the synergism of redox-active V and redox-inert M$^{2+}$ in M$_x$V$_2$O$_5$·nH$_2$O plays a key role in regulating the structure and function of peroxidase mimics, which is similar to the relationship between redox site and Lewis acid in natural peroxidase.

## Excellent antibacterial properties of M$_x$V$_2$O$_5$·nH$_2$O

The enzyme mimics are widely used in antibacterial researches, because the reactive oxygen species (ROS) catalyzed by enzyme mimics can oxidize key components of bacteria, such as cell membranes/walls or intracellular compartments[69-71]. In this work, M$_x$V$_2$O$_5$·nH$_2$O are expected to acquire additional bactericidal ability in the presence of low concentration of medical H$_2$O$_2$ disinfectant due to its high peroxidase-mimicking activity. To verify this, we evaluated the activity of M$_x$V$_2$O$_5$·nH$_2$O against gram-positive *Staphylococcus aureus* (*S. aureus*) and gram-negative *Escherichia coli* (*E. coli*) by standard spread plate method. As shown in Fig. 6A, B, in the presence of low concentration H$_2$O$_2$ (200 μM), M$_x$V$_2$O$_5$·nH$_2$O nanobelts show stronger antibacterial ability toward to *S. aureus* and *E. coli* than V$_2$O$_5$ powder and V$_2$O$_5$ nanobelt. Quantitative analysis shows that the relative survival rate of *E. coli* decreased to 3.63%, 5.19%, and 6.79% after M$_x$V$_2$O$_5$·nH$_2$O and H$_2$O$_2$ treatment, while the relative survival rate of *S. aureus* was 4.37%, 6.65%, and 5.56%, respectively (Fig. 6C, D). In control

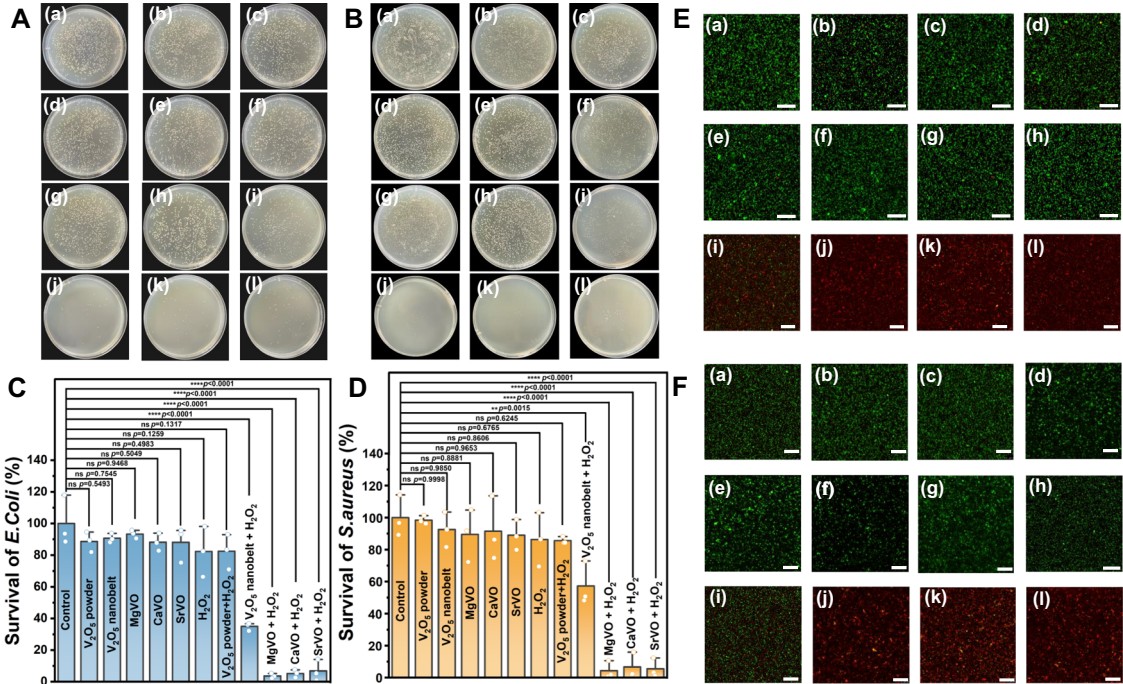

**Fig. 6 | Antibacterial properties of $M_xV_2O_5\cdot nH_2O$.** Photographs of bacterial colonies against *E. coli* (**A**) and *S. aureus* (**B**) with different treatments. Relative survival rate of *E. coli* (**C**) and *S. aureus* (**D**) upon different treatments determined by spread plate method. $n = 3$ biologically independent samples (**C**, **D**). Data are presented as mean values ± SD (**C**, **D**). Data were analyzed by one-way ANOVA with Turkey's multiple comparisons test (**C**, **D**), ns represents no statistical difference,

$*P < 0.01$. $**P < 0.001$. $***P < 0.0001$. $****P < 0.0001$. CLSM images of *E. coli* (**E**) and *S. aureus* (**F**) co-stained with Syto9 and PI after incubating with different treatments. Scale bar: 50 μm. (a) PBS, (b) $V_2O_5$ powder, (c) $V_2O_5$ nanobelt, (d) MgVO, (e) CaVO, (f) SrVO, (g) $H_2O_2$, (h) $V_2O_5$ powder + $H_2O_2$, (i) $V_2O_5$ nanobelt + $H_2O_2$, (j) MgVO + $H_2O_2$, (k) CaVO + $H_2O_2$, (l) SrVO + $H_2O_2$. Representative images are shown from three independent experiments with similar results (**E**, **F**).

experiments without $H_2O_2$ or with $H_2O_2$ alone, the Luria Bertani (LB) agar plate were completely covered with bacterial colonies. This indicates that the bacteria could be inhibited only in the presence of $H_2O_2$ and $M_xV_2O_5\cdot nH_2O$, that is, the antibacterial activity came from the peroxidase-mimicking activity of the materials. To further visually verify the antibacterial ability of the material, SYTO9/PI was used to stain live/dead bacteria. As shown in Fig. 6E, F, compared with other control groups, the red color of PI in the presence of both $H_2O_2$ and $M_xV_2O_5\cdot nH_2O$ was enhanced, indicating a significant antibacterial effect. To further verify the broad-spectrum antibacterial ability of $M_xV_2O_5\cdot nH_2O$, the antibacterial test for gram-positive *Bacillus subtilis* (*B. subtilis*) and gram-negative *Pseudomonas aeruginosa* (*P. aeruginosa*) by standard spread plate method were added. As shown in Supplementary Fig. 29, $M_xV_2O_5\cdot nH_2O$ show significant antibacterial ability against *B. subtilis* and *P. aeruginosa* in the presence of low concentration $H_2O_2$.

## Composite hydrogel with $M_xV_2O_5\cdot nH_2O$ as additive for infected wound healing

In addition to antibacterial properties, the banded structure of $M_xV_2O_5\cdot nH_2O$ are widely believed to contribute to the formation of stable composites. Meanwhile, the large size reduces cytotoxicity due to the difficulty in endocytosis (Supplementary Fig. 30). Therefore, $M_xV_2O_5\cdot nH_2O$ nanobelts are expected to be wound dressing additives for the healing of infected wound. Hence, we then prepared an antibacterial composite hydrogel as wound dressing for wound healing. Concretely, the gelatin solution was mixed with tannic acid solution and the hydrogels (GelTA) rapidly formed via hydrogen bond crosslinking[72]. During the preparation of GelTA, $M_xV_2O_5\cdot nH_2O$ were added to form composite hydrogels for offering wound barrier and promoting the wound healing. Fourier transform infrared (FT-IR) spectrum was used to characterize the prepared hydrogel. The attribution of major characteristic peaks is consistent with relevant reports

(Supplementary Fig. 31)[72]. SEM images showed that freeze-dried GelTA has abundant porous structure with high interconnectivity. $M_xV_2O_5\cdot nH_2O$ functionalized GelTA (Mg-Gel-TA/ Ca-Gel-TA/ Sr-Gel-TA) also displayed similar porous structure, which were beneficial for exchange of substances (Supplementary Fig. 32). EDX elemental quantification shows the existence of $M_xV_2O_5\cdot nH_2O$ in the composite hydrogel. In order to study the peroxidase-mimicking activity of three $M_xV_2O_5\cdot nH_2O$ functionalized GelTA, which were placed in the mixed solution of $H_2O_2$ and TMB. The results showed that the $M_xV_2O_5\cdot nH_2O$ functionalized GelTA could rapidly make the mixed solution turn dark blue, the absorption peak of oxTMB at 652 nm appeared in the UV-vis absorption spectrum (Supplementary Fig. 33), indicating that the $M_xV_2O_5\cdot nH_2O$ functionalized GelTA exhibit peroxidase-mimicking activity, which is expected to have antibacterial ability. In order to verify the antibacterial efficacy of the $M_xV_2O_5\cdot nH_2O$ functionalized GelTA as wound dressings, infected wound healing experiment was carried out on mice. A round whole cortical wound with a diameter of 5.5 mm was formed on the back of each mouse (Supplementary Fig. 34), which was randomly divided into 9 groups for different treatments after 24 h of *S. aureus* infection: (1) PBS, (2) GelTA, (3) GelTA+$H_2O_2$, (4) Mg-GelTA, (5) Ca-GelTA, (6) Sr-GelTA, (7) Mg-GelTA +$H_2O_2$, (8) Ca-GelTA+$H_2O_2$, (9) Sr-GelTA+$H_2O_2$. The wound photos of mice at day 0, 3, 5 and 7 were respectively shown in Fig. 7B. It could be seen that the wound had crusted and no obvious infection occurred on day 3 after treatment with Mg-GelTA+ $H_2O_2$, Ca-GelTA+ $H_2O_2$, Sr-GelTA + $H_2O_2$. On day 5, the relative wound area of mice decreased to 34.1%, 47.8% and 39.9%, which are much smaller than other treatment groups (Fig. 7C). Besides, there were no ulceration or suppuration occurred during the treatment. In contrast, the other 6 groups show slower wound healing and varying degrees of inflammation. The wound healing traces of mice treated with different treatments within 7 days were analyzed (Fig. 7D), which further indicated that the wound healing rate of group (7) (8) (9) are significantly faster than other treatment

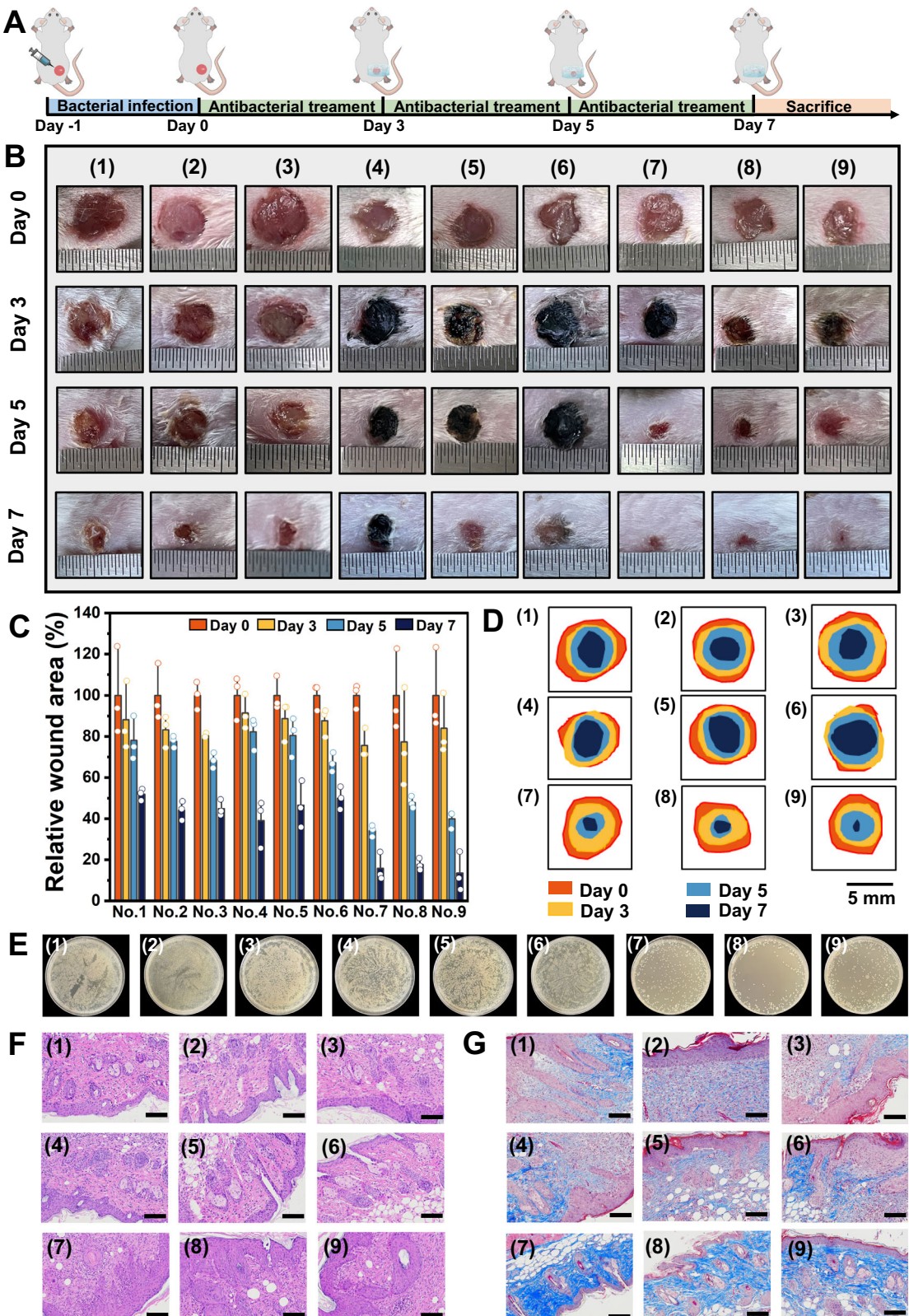

**Fig. 7 | $M_xV_2O_5 \cdot nH_2O$ functionalized hydrogels used as antibacterial dressings to promote wound healing. A** Schematic illustration of the in vivo wound disinfection treatment with wound dressing containing $M_xV_2O_5 \cdot nH_2O$ in a mice model. **B** Photographs of the infected wound on mice and (**C**) relative wound area of different groups at day 0, 3, 5, 7 during antibacterial treatment process. $n$ = 3 biologically independent samples. Data are presented as mean values ± SD. **D** Wound healing traces of mice in each group during 7 days. **E** Photographs of bacterial colonies separated from wound tissue with different treatments at day 5. **F** H&E staining and **G** Masson's trichrome staining of the excised wound tissues treated by different groups. (1) PBS, (2) GelTA, (3) GelTA + $H_2O_2$, (4) Mg-GelTA, (5) Ca-GelTA, (6) Sr-GelTA, (7) Mg-GelTA + $H_2O_2$, (8) Ca-GelTA + $H_2O_2$, (9) Sr-GelTA + $H_2O_2$. Scale bar, 100 μm. Representative images are shown from three independent experiments with similar results (F, G).

group. In addition, the bacteria from wound tissues of mice in each group were cultured on LB agar plates, and the results showed that the number of bacteria significantly decreased at the wound sites in the treatment groups (Fig. 7E, Supplementary Fig. 35). Further, Hematoxylin and eosin (H&E) staining was analyzed for the wound tissue of different groups of mice after seven days of treatment (Fig. 7F). The results show that the epidermal layers at the wound sites are more intact in the treatment groups, and only a few inflammatory cells are distributed. In the control groups, the epidermal layers at the wound sites are fragmented and many inflammatory cells gather in the wound areas. Masson staining shows that the density of collagen fibers at the wound site are higher in the treatment groups than in the control groups (Fig. 7G). To ensure its biosafety in vivo, weight changes in mice were recorded during the treatment. As shown in Supplementary Fig. 36, there are no significant changes in body weight the body weight of mice during the treatment, indicating that the mice lived well. Further, H&E staining analysis of the major organs of the mice in all experimental groups show that no obvious organ damage, abnormality or inflammation occurred in the mice (Supplementary Fig. 37). In conclusion, $M_xV_2O_5 \cdot nH_2O$ functionalized GelTA can effectively inhibit bacterial growth and promote wound healing as a biosafe antibacterial wound dressing.

In summary, we have developed a series of $M_xV_2O_5 \cdot nH_2O$ peroxidase mimics with synergistic structure of redox site and Lewis acid, which have more significant peroxidase-mimicking activity compared with other typical peroxidase mimics according to the $V_{max}$ and TON values comparison. The experimental results and theoretical calculation indicate that the prominent enzyme-mimicking catalytic activity of $M_xV_2O_5 \cdot nH_2O$ are caused by the change of geometric and electronic structure after redox-inert $M^{2+}$ intercalation. Specifically, the $M^{2+}$ intercalation lead to the central redox-active V in $V_2O_5$ crystal transform into a low coordination number structure, and the resulting high concentration of oxygen vacancies can initially reduce the energy of $H_2O_2^*$ dissociation. More importantly, the charge redistribution around V caused by $M^{2+}$ intercalation makes V more incline to connect with $OH^*$, thus promoting the $H_2O_2^*$ dissociation to become completely spontaneous. This significantly enhanced the POD-mimicking activity of $M_xV_2O_5 \cdot nH_2O$. Interestingly, we found that the catalytic activity of $M_xV_2O_5 \cdot nH_2O$ are periodically related to the Lewis acidity of intercalated $M^{2+}$, it should be caused by the non-covalent polarization of intercalated metal ions. The redox-inert $M^{2+}$ regulation of the geometrical and electronic structure of the $V_2O_5$ host highly mimics the synergistic effect of the redox site and Lewis acid in natural peroxidase. On this basis, $M_xV_2O_5 \cdot nH_2O$ were applied to antibacterial therapy as an antibacterial additive in wound dressing. In vivo and in vitro experiments showed that $M_xV_2O_5 \cdot nH_2O$ functionalized wound dressings displayed significant antibacterial activity in the presence of low concentration of medical hydrogen peroxide. This study not only reported a series of highly efficient peroxidase mimics with practical application potential, but also proposed and explained a strategy to activate enzyme-mimicking activity through synergetic structure of redox site and Lewis acid at the experimental and theoretical levels. At the same time, it also provides a general idea for the regulation of enzyme-mimicking catalytic activity of many other metal oxides layered compounds.

## Methods

### Ethical statement
All animal experiments were carried out in compliance with the 3 R principle, and were approved by Animal Ethics Committee of Shandong Normal University under approval number of AEECSDNU2022100.

### Materials
3,3′,5,5′-tetramethylbiphenamine (TMB), riboflavin and 5,5-dimethyl-1-pyrrolin-n-oxide (DMPO) were purchased from Sigma Aldrich Trading Co., Ltd. Magnesium chloride hexahydrate ($MgCl_2 \cdot 6H_2O$), Calcium chloride ($CaCl_2$), strontium cholride hexahydrate ($SrCl_2 \cdot 6H_2O$), vanadium pentoxide ($V_2O_5$), Polyethylene glycol-4000 (PEG-4000) and dimethyl sulfoxide (DMSO) were purchased from Shanghai Macklin Biochemical Co., Ltd. Sodium acetateanhydrous, acetic acid, isopropanol, and hydrogen peroxide (wt 30%) were purchased form Sinopharm Chemical Reagent Co., Ltd. Nitro blue tetrazolium (NBT), L-Methionine, 1,2-diaminobenzene and 2,2′-azino-bis(3-ethylbenzothiazoline-6-sulfonic acid) (ABTS) were purchased from Shanghai Aladdin Biochemical Technology Co., Ltd. SYTO 9/PI Live/Dead Bacterial Double Stain Kit was purchased from Shanghai Maokang Biotechnology Co., Ltd. E. coli (ATCC 25922), B. subtilis (ATCC 6051), S. aureus (ATCC 6538) and P. aeruginosa (ATCC 27853) were purchased from Beijing Solarbio Science & Technology Co., Ltd.

### Preparation of $M_xV_2O_5 \cdot nH_2O$ nanobelts
The $M_xV_2O_5 \cdot nH_2O$ (M = Mg, Ca, Sr) nanobelts were synthesized by using an one-step hydrothermal method[38]. $Mg_{0.22}V_2O_5 \cdot 0.75H_2O$ (MgVO) was synthesized by following steps: First, 2 mmol $V_2O_5$ was dispersed to the 54 mL deionized water under stirring. After stirring for about 30 min, 10 mL $H_2O_2$ (30 wt%) was then slowly added into the above solution with vigorous stirring until the orange-red solution was obtained. Then 0.04 g PEG-4000 and 60 mmol $MgCl_2 \cdot 6H_2O$ was added into the solution, and stirred for 3 h at room temperature, the transparent, orange-red solution turned into a dark-red solution. The final solution is transferred into a 100 mL Teflon-lined stainless steel autoclave and hydrothermally treated at 200 °C for 48 h. The autoclave was cooled to room temperature, and the obtained products was washed with water and ethanol for three times. Finally, dark green MgVO nanobelts were obtained by vacuum dried at 60 °C for 10 h.

$Ca_{0.30}V_2O_5 \cdot 0.83H_2O$ (CaVO) and $Sr_{0.34}V_2O_5 \cdot 0.72H_2O$ (SrVO) nanobelts were also synthesized with the above method. When CaVO nanobelts were prepared, $MgCl_2 \cdot 6H_2O$ (60 mmol) were replaced by $CaCl_2$ (60 mmol) and dispersed in 60 mL deionized water, and other conditions remain unchanged.

Likewise, when SrVO nanobelts were prepared, $MgCl_2 \cdot 6H_2O$ (60 mmol) were replaced by $SrCl_2 \cdot 6H_2O$ (60 mmol), and other conditions remain unchanged.

### Assay of peroxidase-mimicking activity of $M_xV_2O_5 \cdot nH_2O$ nanobelts
The peroxidase-mimicking activity of $M_xV_2O_5 \cdot nH_2O$ nanobelts were determined by measuring the oxidation of TMB in the presence of $H_2O_2$. $M_xV_2O_5 \cdot nH_2O$ nanobelts (5 μg/mL) were added into 2 mL HAc-NaAC buffer (0.2 M, pH = 5.0) containing TMB (2 mM) and $H_2O_2$ (200 mM). The absorption intensities of oxidized TMB (oxTMB) at 652 nm were monitored by a UV-vis spectroscopy. In addition, the peroxidase-mimicking activity of $M_xV_2O_5 \cdot nH_2O$ nanobelts was further determined by ABTS and OPD instead of TMB.

### Electrochemical detection
First, glassy carbon electrodes (GCEs) were polished using alumina oxide powder (0.05 μm) and then washed with deionized water. Then the 20 μL mixed solution containing $M_xV_2O_5 \cdot nH_2O$ nanobelts (1 mg/mL) and Nafion (10% in alcohol) was drop on the GCE to obtain the $M_xV_2O_5 \cdot nH_2O$ modified GCEs. Cyclic voltammetry (CV) were carried out to monitor the catalytic activity of the $M_xV_2O_5 \cdot nH_2O$ modified GCEs in 3 M KCl with different concentrations of $H_2O_2$ (0.1–0.5 mM) and fixed concentrations of TMB (0.2 mM).

### Steady-state kinetic assays
The Steady-state kinetic assays were carried out in a 2.0 mL cuvette (optical path l: 5 mm) containing $M_xV_2O_5 \cdot nH_2O$ nanobelts (5 μg/mL) in 2 mL HAc-NaAc buffer (0.2 M, pH = 5.0). To obtained kinetic data toward for TMB, the different concentrations of TMB (0.2-3 mM) and

the fixed concentrations of $H_2O_2$ (200 mM) were added to the 2 mL HAc-NaAc buffer (0.2 M, pH = 5.0) containing $M_xV_2O_5 \cdot nH_2O$ nanobelts (5 µg/mL). Similarly, to obtained kinetic data toward for $H_2O_2$, the different concentrations of $H_2O_2$ (2–300 mM) and the fixed concentrations of TMB (2 mM) were added to the 2 mL HAc-NaAc buffer (0.2 M, pH = 5.0) containing $M_xV_2O_5 \cdot nH_2O$ nanobelts (5 µg/mL). The absorbance intensities at 652 nm was measured by a UV spectroscopy as a function of time with time intervals of 1 s for 1 min. The initial reaction rate within 5 s was calculated according to the following formula: $v = \Delta A / \Delta t (\varepsilon_{652} \times l)$, where $\varepsilon_{652}$ is the molar absorption coefficient of oxidized TMB at 652 nm ($\varepsilon_{652} = 3.9 \times 10^4$ $M^{-1} \cdot cm^{-1}$), and $l$ is the optical path of cuvette ($l = 5$ mm). The kinetic parameter was derived from the Michaelis-Menten equation: $v = (V_{max} \times [S]) / (K_m + [S])$, where v is the initial velocity, [S] is the concentration of the substrate, $K_m$ is the Michaelis−Menten constant, and $V_{max}$ is the maximal reaction velocity. $TON = V_{max}/[E]$, where [E] is the molar concentration of MgVO/CaVO/SrVO.

## DFT calculation

The spin-polarized density functional theory (DFT) calculations have been conducted on Vienna ab-initio simulation package (VASP)[73,74] to study the alkaline hydrogen evolution reaction. The Projector augmented wave method[75] with a cutoff energy of 450 eV accompanied by Perdew-Burke-Ernzerhof functional[75] has been used in the DFT calculations. DFT + U method[76] with the effective U value of 3.0 eV[77] and DFT-D3 method[78] was used to correct the influence of 3d electrons of V atom and van der Waals interactions, respectively. Two layers of $V_2O_5$ (001) facet have been cleaved with a vacuum layer of 15 Å to build the $V_2O_5$ slab model and half of bottom layers have been fixed to simulate the bulk phase. Four of Mg-atom, Ca-atom and Sr-atom with 16 of $H_2O$ molecules have been inserted in the interlamination of $V_2O_5$, respectively to build the MgVO, CaVO and SrVO models. Besides, 16 of $H_2O$ molecules have been inserted in the interlamination of $V_2O_5$ to build the $V_2O_5 \cdot nH_2O$ model. All models have been fully relaxed with the energy convergence criterion of $10^{-5}$ eV and the force convergence criterion of 0.02 eV/Å, respectively. The free energy of $H^+$ is obtained by $H^+ + e^- \rightarrow 0.5\, H_2$ according to the National Institute of Standards and Technology chemistry webbook and the free energy of TMB-$H^+$ has been calculated by regarding TMB-$H^+$ as a radical. The $(1 \times 2 \times 1)$ K-points have been used as K-point mesh. The adsorption energy ($E_{ads}$) has been calculated using formula 2,

$$E_{ads} = E_{total} - E_{substrate} - E_{adsorbate} \qquad (2)$$

The $E_{total}$, $E_{substrate}$ and $E_{adsorbate}$ represent the energy of adsorption structure, substrate and adsorbate, respectively. The free energies have been calculated using the following formula 3,

$$G = E_{DFT} + ZPE - TS \qquad (3)$$

The G, $E_{DFT}$, ZPE and TS represent the free energy, energy from DFT calculations, zero point energy and entropic contributions, respectively.

## The deintercalation on the activity of $M_xV_2O_5 \cdot nH_2O$

MgVO, CaVO and SrVO were respectively placed in deionized water and the solution is adjusted to pH = 10.0 with 1 M NaOH. The above solution was heated under nitrogen for 8 h. Then the products were collected and alternately cleaned with ethanol and water for three times, finally, the MgVO, CaVO and SrVO after deintercalation were obtained by vacuum dried at 60 °C for 10 h. The peroxidase-mimicking activity of $M_xV_2O_5 \cdot nH_2O$ before and after deintercalation under the same condition with 2 mL HAc-NaAC buffer (0.2 M, pH = 5.0) containing TMB (2 mM) and $H_2O_2$ (200 mM).

## In vitro antibacterial activity

The antibacterial activity of nanozymes were evaluated by a spread plate method. In brief, *E. coli* (ATCC 25922) or *S. aureus* (ATCC 6538) were respectively diluted with sterilized HAc-NaAc buffer (0.2 mM, pH = 5.0) to $10^5$ CFU·mL$^{-1}$ and added to 24-well cell culture plates. Then $V_2O_5$ powder, $V_2O_5$ nanobelt, $M_xV_2O_5$ (MgVO/CaVO/SrVO) with final concentration of 100 µg/mL and $H_2O_2$ (200 µM) were added, respectively. The total volume of solution in each well was 500 µL. After incubation at 37 °C and 180 rpm for 30 min, 100 µL bacterial suspension was taken out and evenly coated on LB agar plate. The survival bacterial colonies were present on the plates by culturing at 37 °C for 12 h.

## Live/dead fluorescent staining of bacteria

A confocal laser microscope was used to observe the survival and death status of bacteria. The bacterial suspension (*E. coli* (ATCC 25922) or *S. aureus* (ATCC 6538), $10^5$ CFU mL$^{-1}$) was respectively diluted by sterilized HAc-NaAc buffer (0.2 mM, pH = 5.0), then the bacterial suspension was incubated with 100 µg/mL $V_2O_5$ powder, $V_2O_5$ nanobelt, $M_xV_2O_5 \cdot nH_2O$(MgVO/CaVO/SrVO) and 200 µM $H_2O_2$ at 37 °C at 180 rpm for 4 h. Finally, the bacteria were stained using the SYTO 9/PI live/dead bacterial double stain kit.

## Cytotoxicity validation of $M_xV_2O_5 \cdot nH_2O$

The effect of $M_xV_2O_5 \cdot nH_2O$ on cell viability was determined using CCK8 (Beyotime Biotech, China). HMEC-1 (CRL-3243) and L929 (CCL-1) cells were acquired commercially form ATCC, which were authenticated by STR. Briefly, HMEC-1or L929 cells were digested into suspension and plated in 96-well plates with 5000 cells per well for 24 h incubation. Then, these cells were cocultured with various concentrations (20-300 µg/mL) of $M_xV_2O_5 \cdot nH_2O$ for 8 h. After the cells were washed, 100 µL of CCK8 dilution was added, and the cells were incubated for some time to detect the absorbance of cells at 450 nm.

## Preparation of GelTA hydrogel

GelTA hydrogels were prepared by hydrogen bonding between gelatin and tannic acid (TA) using the previously reported method[72]. Gelatin is dissolved by stirring in deionized water at 70 °C to obtain 10% w/v gelatin solution. Then, 0.5 mg/mL TA solution was prepared in deionized water. 120 µL TA solution is added drop by drop to 1 mL 10% w/v gelatin solution and stirred to obtain GelTA hydrogel. Finally, the GelTA hydrogel was freeze-dried for later using.

## Preparation of wound dressing based on $M_xV_2O_5 \cdot nH_2O$

100 mg gelatin were added to 0.9 mL deionized water and heat it to dissolve at 70 °C, then 500 µL 1 mg·mL$^{-1}$ MgVO/CaVO/SrVO were respectively added to the gelatin solution, finally, 120 µL 0.5 mg·mL$^{-1}$ TA solution was added to the mixed solution, and the $M_xV_2O_5 \cdot nH_2O$ functionalized GelTA hydrogels were obtained by stirring the solution as wound dressing. After being freeze-dried, a round wound dressing with uniform size (5.5 mm diameter) was obtained by a hole punch.

## In vivo Wound healing assessment with *S. aureus* infection

All animal experiments were approved by the Animal Ethics Committee of Shandong Normal University (AEECSDNU2022100). We established a full-thickness wound model caused by *S. aureus* (ATCC 6538) on the back of female Balb/c mice (6 weeks, wide-type, Jinan Pengyue Laboratory Animal Breeding Co., LTD., China). This study only included female mice, and we did not pay attention to the effect of gender on wound healing. Since male mice are more prone to bite wounds and dressings than female mice when fed in the same cage, we chose more docile female mice. All mice fed in a barrier environment and mice were kept in constant temperature (22°C), constant humidity (55%) and cyclic lighting (12 h light/12 h dark). Prior to surgery, mice were anesthetized with intraperitoneal injection of 100 µL chloral hydrate

(3.5%). After removing the back hair of mice, the 5.5 mm diameter full-thickness wounds were established with surgical scissors. 50 μL *S. aureus* (ATCC 6538) suspension ($1 \times 10^8$ CFU mL$^{-1}$) were added to the wound sites, after 24 h infection, mice were randomly divided into nine groups and different treatment were administered to the wound sites of the nine groups and each group had an average of three mice: (1) PBS, (2) GelTA, (3) GelTA+$H_2O_2$, (4) Mg-GelTA, (5) Ca-GelTA, (6) Sr-GelTA, (7) Mg-GelTA+$H_2O_2$, (8) Ca-GelTA+$H_2O_2$, (9) Sr-GelTA+$H_2O_2$. The concentration of $H_2O_2$ was the same as in vitro antibacterial test. The wound sites of mice were treated every 24 h, different wound dressings were fixed on the wound sites with medical gauze, and the wound dressings were removed after 1 h. The body weight of mice was recorded, and the wound area was measured and photographed to record the wound healing process at day 0, 3, 5 and 7, respectively. The wound dressings were changed when the wound areas were being photographed.

The number of bacteria at the wound sites were measured by spread plate method, mice with different treatment were sacrificed on day 7 and the related wound tissues were harvested and placed in normal saline. The obtained bacterial suspension was spread on LB agar plates and incubated at 37 °C for 18 h.

### Histological and immunofluorescence analysis

After 7 days of treatment, mice (female Balb/c, 6 weeks, wide-type) in each group were sacrificed, and the related wound tissue was fixed in 4% paraformaldehyde for 24 h. Subsequently, wound tissues were embedded in paraffin, sectioned, stained with hematoxylin and eosin (H&E) staining and Masson's trichrome staining to evaluate the wound healing effect. All samples were examined with a microscope.

### Reporting summary

Further information on research design is available in the Nature Portfolio Reporting Summary linked to this article.

## Data availability

All data generated in this study are provided in the Supplementary Information/Source Data file. Source data are provided with this paper.

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

## Acknowledgements

This work was supported by National Natural Science Foundation of China (22134004 to B.T., 22074082 to L.L.), Natural Science Foundation of Shandong Province of China (ZR2023MB050 to H.S.), Key R&D Plan of Shandong Province (2021ZDPT01 to L.L.), Taishan Scholars Program of Shandong Province (tsqn.201909077 to L.L.).

## Author contributions

These authors contributed equally: Haibin Si, Dexin Du. The project was conceptually designed by Bo Tang, Lu Li and Haibin Si. The majority of the experiments were conducted by Dexin Du and Haibin Si (with equal contributions), assisted by Chengcheng Jiao and Yan Sun. Data analysis and interpretation was done by Dexin Du. The manuscript was prepared by Haibin Si, Dexin Du and Lu Li. All authors have given approval to the final version of the manuscript.

## Competing interests

The authors declare no competing interests.
