## [Peer Review File · Nature Communications]

Reviewers' Comments:

Reviewer #1:

Remarks to the Author:

The authors of this manuscript have developed a series of alkaline earth metal (Ca, Mg, and Sr) intercalated V₂O₅ nanobelts as peroxidase mimics. This study was inspired by the role of alkaline earth metals present in the peroxidase enzyme, assisting in its activity. They have demonstrated that intercalation-induced geometric and electronic alterations affect vanadium centers, leading to the generation of high concentrations of oxygen vacancies. This effect enhances H₂O₂ activation to OH radicals, thereby exhibiting antibacterial activity. Consequently, this activity aids in wound healing in mice. The authors demonstrate the peroxidase-mimicking activity of MxV₂O₅·nH₂O, showing a periodic change with the Lewis acidity (ion potential) of Ca, Mg, and Sr and this work has some potential. However, authors should consider other reports in the literature that have demonstrated the ability of alkaline earth metals, in combination with transition metals, to mimic peroxidase activity (see references: <https://doi.org/10.1016/j.snb.2022.131869>, <https://doi.org/10.1016/j.mattod.2023.06.015>, <https://doi.org/10.1016/j.mtchem.2023.101861>, <https://doi.org/10.3390/chemosensors9080219>). Although the current manuscript also deals with the antibacterial and wound-healing activities of their nanozymes, there exist numerous articles on this topic. This manuscript introduces some novel aspects, such as enhancing the activities of V₂O₅ nanobelts by non-redox metal intercalation. However, there are several issues to be addressed for improving this manuscript, as discussed below:

1. What is the origin of the peak observed near $2\theta = 10^\circ$ in the XRD pattern?
2. Although the authors have provided the length and width of the nanobelts, the thickness remains unspecified, which is crucial for defining them as nanobelts. It is recommended that the authors employ appropriate techniques such as AFM to characterize the structures thoroughly.
3. Despite the authors' indication of a width ranging from 200-300 nm based on SEM images, the TEM image distinctly reveals a width exceeding 500 nm. It is advised to review the TEM and EDX mapping images.
4. The key challenges faced by nanozymes are selectivity and specificity. These aspects are currently under active investigation by researchers working in the artificial enzyme area. While the authors assert that their nanozymes exhibit excellent and specific peroxidase-mimicking activity compared to oxidase, catalase, and SOD, previous studies suggest that various vanadium-based systems like nanomaterials and metal-organic frameworks also demonstrate haloperoxidase and glutathione peroxidase activities. Furthermore, there are reports highlighting the efficacy of highly efficient peroxidase nanozymes (<https://www.nature.com/articles/s41467-022-32411-z>).
5. While most nanozymes exhibit peroxidase-like activity through oxidation by OH radicals, this differs from natural peroxidase enzymes, which utilize ferryl oxo species for oxidation instead of OH radicals.
6. The V_{max} and K_M value bars should be presented with error bars. Additionally, it should be noted that there are significant errors in evaluating the kinetic parameters. For instance, the V_{max} for SrVO nanobelts is indicated as $71.7 \times 10^{-8} \text{ M}\cdot\text{s}^{-1}$ (see supporting information Figure 9B(b)); however, this value is notably lower than the plateau region in the plot, which is not possible. Similarly, K_M should ideally reflect the concentration at half of the actual V_{max}, and these values significantly differ in most of the kinetic plots. Authors should carefully examine these aspects as these values are used for comparison with other reported nanozymes.
7. Authors have compared the activities of other peroxidase mimics, but the basis of comparison remains unclear. Considering the differences in morphologies, sizes, and structures among these systems, it becomes challenging to make direct comparisons. Utilizing the quantity of materials while maintaining a constant surface area for all samples may be more relevant in this case.
8. In Figure 4a, hydrogen peroxide should be adsorbed and not absorbed. This correction should be applied throughout the figure and text. Moreover, in step 2 of this figure, the term "hemolysis" used by the authors requires clarification. After step 3, the OH radical is depicted near the surface of the material, but its reaction with a proton (which should be proton adsorption) results in a water molecule. If this is the case, the generated water molecule would lack one electron, which is not feasible. Authors should carefully review the mechanism.

9. Authors should also furnish evidence regarding the oxidation states of metal centers during catalysis. Is there any change in the oxidation state when TMB interacts with the catalyst and H₂O₂?
 10. In line 398, Authors say "the proposed MxV₂O₅·nH₂O were extremely similar in structure and function to natural peroxidase". On what basis are they extremely similar in structure? The mechanism is also quite distinct from natural enzyme (please see comment no. 5).
 11. The name of the bacteria should always be conventionally written in italics. For example, *Staphylococcus aureus*. Please correct such mistakes.
 12. Authors have only used two bacteria in this study, but they claim that their nanozymes have broad-spectrum activity. Evaluation should be conducted using more strains to substantiate this claim.
 13. I see that the nanozymes demonstrate antibacterial activities and not bacteriostatic activities. Kindly correct it in line 429.
 14. In Fig 6B, the dimensions of the wound (Day 0) in lanes 1, 2, and 3 appear to be different from the others. How can these be correctly compared in the study?
 15. In Supplementary Figure 18, no belt-like structures are visible in images B, C, and D. Considering that these belts have lengths of 1-4 μm, they are not visible at the scale size of 5 and 10 μm in the provided SEM images.
 16. Authors have compared their nanozymes' activities with HRP and Fe₃O₄ reported by Gao et al. However, this comparison may not be appropriate as HRP and Fe₃O₄ have been evaluated under different reaction conditions. Therefore, it is improper to conclude that "the peroxidase-mimicking activity of MxV₂O₅·nH₂O was more than one order of magnitude higher".
 17. There are grammatical errors in multiple places.
- Considering all the points raised above, the manuscript should be thoroughly improved addressing above point for its acceptance in Nature Communications.

Reviewer #2:

Remarks to the Author:

Lewis acids such as Ca²⁺ play multifaceted roles in biological enzymes, especially in the catalytic action of peroxidases, including stabilizing the active site structure, promoting substrate binding, regulating the catalytic process, stabilizing the enzyme-substrate complex, and adjusting enzyme activity. This provides a good perspective for the rational design of nanozymes, but current research on nanozyme designs that consider the auxiliary role of Lewis acids is limited, so the creativity of this paper is innovative. However, there are still some issues that need improvement, and I suggest considering publication in Nature Communications after major revision. Here are my suggestions:

1. Please clarify the long-term stability of the material under acidic conditions. Does the material degrade?
2. The theoretical calculation part of the mechanism does not consider the substrate, such as TMB. The adsorbed two HO* are directly desorbed, and one combines with H⁺ in the solution to get an electron from the material to form water and desorb. I don't think this mechanism is very reasonable because: a. The relative activity of the nanozyme is related to the substrate, so the substrate should be considered. b. The calculation results of this paper show that the adsorption of HO* is an endothermic process, so the free •OH adsorption on the surface is a spontaneous exothermic process. Therefore, compared to the mechanism of releasing free radicals, I think the process of continuously providing reductive hydrogen [H] to the two adsorbed HO* by reductive substrates, such as TMB, reported in the previous literature (Advanced Materials 2023, 2211151.) is more reasonable.
3. The adsorption energy of H₂O₂ for noble metals, metal oxides, carbon-based materials, etc., is roughly between 0.0 and -1.0. However, the calculated adsorption energy of H₂O₂ in this paper is very strong, reaching -4.0 to -5.0 eV, which has obviously exceeded the strength of chemical adsorption, indicating that new chemical bonds have been formed. Please confirm the accuracy of the calculation, explain the reasons, or are there any literature reports of similar results?

4. There are already many theoretical reports on the mechanism of POD activity of nanozymes, but the authors rarely cite and discuss them.
5. Why didn't the authors consider the effect of surface Ca^{2+} , instead of interlayer Ca^{2+} , on enzyme activity?
6. Using Lewis acid assistance to improve enzyme activity and perform structure-activity relationship analysis is an interesting topic, but the authors did not clearly explain the actual impact of Lewis acid on VO materials, simply explaining it as the polarization effect of ionic potential on VO. The effect from electronic effects is reasonable, but it should be thoroughly explained. I suggest that the authors calculate the band structure of the models and pay attention to the frontier energy levels, and discuss the activity in combination with the redox potential, because the essence of the substrate [H] snatching is a redox reaction. Therefore, the impact of Lewis acid on activity can be discussed from the perspective of the influence of electronic effects on the redox potential, which can be done by comparing the differential charge density analysis before and after addition, rather than simply analyzing the electron transfer of OH^* , because there is no significant difference in the results of figure 4F, c, d, e as it stands.
7. Line 324-325, the authors did not calculate the transition state, so I don't understand the term "barrier of decomposition", and why the barriers are all negative, which seems inconsistent with the results in the figure.
8. How did the authors calculate the charged system, by adding a background charge? Please supplement the explanation in the calculation method.

Reviewer #3:

Remarks to the Author:

This manuscript reported a series of $\text{MxV}_2\text{O}_5 \cdot n\text{H}_2\text{O}$ peroxidase mimics with vanadium as the redox site and redox-inert alkaline-earth metal ion (M^{2+}) as the Lewis acid, which have more significant peroxidase-mimicking activity compared with other typical peroxidase mimics according to the maximal reaction velocity (V_{max}) the turnover number (TON). The mechanism has been analyzed carefully by experiments and theoretical calculations. On this basis, $\text{MxV}_2\text{O}_5 \cdot n\text{H}_2\text{O}$ was used for antibacterial treatment and achieved remarkable effect. But some shortcomings also should be further corrected in the manuscript. I prefer to accept it after major revision. Some comments are listed as follows:

1. The quality of Fig 1C and D should be improved. Spherical aberration electron microscope is recommended.
2. The biological safety of materials is very important, and it is suggested that the authors examine the safety of materials on normal cells, such as skin cells, vascular endothelial cells, fibroblasts, and so on. These experiments must be rigorous and substantial.
3. I can not identify $\text{MxV}_2\text{O}_5 \cdot n\text{H}_2\text{O}$ nanomaterials in SEM images in Fig. S18. And the quality of SEM images of hydrogels were poor.
4. The bacteria in the wound sites of different treatment groups need to be further characterized.
5. What is the mechanism of material sterilization? Can hydroxyl radicals cross bacterial cell walls? Bacteria are more resistant to free radicals than mammalian cells due to the protection of cell wall. If the free radicals produced by the material are able to kill the bacteria, could the damage to the surrounding normal skin cells be greater? How to avoid damage to the normal cells of the wound? In addition, free radicals can cause wound inflammation. How to avoid the wound inflammation caused by free radicals?

REVIEWER COMMENTS

Reviewer #1 (Remarks to the Author):

The authors of this manuscript have developed a series of alkaline earth metal (Ca, Mg, and Sr) intercalated V_2O_5 nanobelts as peroxidase mimics. This study was inspired by the role of alkaline earth metals present in the peroxidase enzyme, assisting in its activity. They have demonstrated that intercalation-induced geometric and electronic alterations affect vanadium centers, leading to the generation of high concentrations of oxygen vacancies. This effect enhances H_2O_2 activation to OH radicals, thereby exhibiting antibacterial activity. Consequently, this activity aids in wound healing in mice. The authors demonstrate the peroxidase-mimicking activity of $M_xV_2O_5 \cdot nH_2O$, showing a periodic change with the Lewis acidity (ion potential) of Ca, Mg, and Sr and this work has some potential. However, authors should consider other reports in the literature that have demonstrated the ability of alkaline earth metals, in combination with transition metals, to mimic peroxidase activity (see references: <https://doi.org/10.1016/j.snb.2022.131869>, <https://doi.org/10.1016/j.mattod.2023.06.015>, <https://doi.org/10.1016/j.mtchem.2023.101861>, <https://doi.org/10.3390/chemosensors9080219>). Although the current manuscript also deals with the antibacterial and wound-healing activities of their nanozymes, there exist numerous articles on this topic. This manuscript introduces some novel aspects, such as enhancing the activities of V_2O_5 nanobelts by non-redox metal intercalation. However, there are several issues to be addressed for improving this manuscript, as discussed below:

Response: Thank you very much for your suggestions. With respect to "However, authors should consider other reports in the literature that have demonstrated the ability of alkaline earth metals, in combination with transition metals, to mimic peroxidase activity", alkaline earth metal ions as common cations may appear in many materials for charge balance or as templates/substrates. But so far, most of the enzyme mimics have focused on redox-active transition metals, and the role of redox-inert alkaline earth metals in enzyme-like catalysis have not been received much attention. The synergistic effect of transition metals and alkaline earth metals is also not proposed. We have carefully read and analyzed above references. In these papers, the simulated enzymes are $SrMnO_3$, $SrRuO_3$, $SrFeO_3$ (Sens. Actuators, B, 2022, 364, 131869), calcium hexacyanoferrate (III) nanoparticles (Mater. Today, 2023, 68, 148), FeCaP nanoparticles (Mater. Today Chem., 2024, 35, 101861) and Mg-Aminoclay-Based Fe_3O_4/TiO_2 Hybrids (Chemosensors, 2021, 9, 219). The focus of these literatures lies in the application of Fe/Mn/Ru-based enzyme mimics, and there is no detailed exploration of structure-activity relationships. Importantly, alkaline earth metals exist in these materials just for charge balance ($SrMnO_3$, $SrRuO_3$, $SrFeO_3$) and as substrates (calcium hexacyanoferrate (III) nanoparticles, FeCaP nanoparticles and Mg-Aminoclay-Based Fe_3O_4/TiO_2 Hybrids), and there is no further experimental validation or discussion about the role of alkaline earth metals in enzyme-like catalysis in these literatures.

We sincerely appreciate your recognition and comments on our work. The point-to-point responses to your comments are as follows:

1. What is the origin of the peak observed near $2\theta = 10^\circ$ in the XRD pattern?

Response: The diffraction peaks near $2\theta = 10^\circ$ are attributed to the (001) crystal plane. The sharp diffraction peak of $M_xV_2O_5 \cdot nH_2O$ near $2\theta = 10^\circ$ indicates that the crystal displays lamellar ordering as dominated by the pronounced (001) reflections (Chem, 2019, 5, 1194). With the intercalation of M^{2+} and H_2O in $M_xV_2O_5 \cdot nH_2O$, the lattice spacing of (001) increases, causing the diffraction peak of (001) shift to a smaller angle (Angew. Chem. Int. Ed., 2018, 57, 3943). By introducing the angle of diffraction peak into the Bragg equation $2d\sin\theta = n\lambda$, the lattice spacing of (001) in $M_xV_2O_5 \cdot nH_2O$ can be obtained. We have added the relevant description in the revised manuscript.

The manuscript was modified as follows:

Page 6:

In particular, compared with V_2O_5 powder and V_2O_5 nanobelt, sharp diffraction peaks before $2\theta = 10^\circ$ appeared in $M_xV_2O_5 \cdot nH_2O$ (Fig. 1F, Supplementary Fig. 6). The diffraction peaks near $2\theta = 10^\circ$ are generally attributed to the (001) crystal plane, which indicate that the crystal displays lamellar ordering as dominated by the pronounced (001) reflections⁴⁰. With the intercalation of M^{2+} and H_2O in $M_xV_2O_5 \cdot nH_2O$, the lattice spacing of (001) increases, causing the diffraction peak of (001) shift to a smaller angle⁴¹.

2. Although the authors have provided the length and width of the nanobelts, the thickness remains unspecified, which is crucial for defining them as nanobelts. It is recommended that the authors employ appropriate techniques such as AFM to characterize the structures thoroughly.

Response: Thank you very much for your reminding. Although we preliminarily judged from SEM images that $M_xV_2O_5 \cdot nH_2O$ have typical belt-like structure, we did not perform a rigorous characterization of them. In the revised manuscript, we added the AFM characterization of $M_xV_2O_5 \cdot nH_2O$ (Supplementary Fig. 4). The results show that the thicknesses of most $M_xV_2O_5 \cdot nH_2O$ materials are in the range of 20-60 nm, and significantly less than the widths of these materials. Due to their large aspect ratio and nanoscale thickness, the $M_xV_2O_5 \cdot nH_2O$ materials are reasonably defined as $M_xV_2O_5 \cdot nH_2O$ nanobelts.

The manuscript was modified as follows:

Page 4:

In addition, atomic force microscope (AFM) images show that the thicknesses of most $M_xV_2O_5 \cdot nH_2O$ materials are in the range of 20-60 nm (Supplementary Fig. 4). Due to their large aspect ratio and nanoscale thickness, the $M_xV_2O_5 \cdot nH_2O$ materials are also defined as $M_xV_2O_5 \cdot nH_2O$ nanobelts.

The data were added in supporting information:

Supplementary Fig. 4 AFM images (left) with height profile (right) of MgVO (A), CaVO (B) and SrVO (C). (a), (b) are two representative materials, respectively.

3. Despite the authors' indication of a width ranging from 200-300 nm based on SEM images, the TEM image distinctly reveals a width exceeding 500 nm. It is advised to review the TEM and EDX mapping images.

Response: We are very sorry that the previous data are inaccurate. We used SEM to remeasure the width of $M_xV_2O_5 \cdot nH_2O$, and added the statistical results in Supplementary Fig. 3. According to the statistical results, the widths of $M_xV_2O_5 \cdot nH_2O$ are mainly distributed in the range of 200 ~ 450 nm. The corresponding data have been changed in the revised manuscript. In addition, the TEM and EDX mapping images of $M_xV_2O_5 \cdot nH_2O$ were replaced with more representative ones (Fig.1B, E). In order to provide a more comprehensive view of the morphology and size of these materials, we also added TEM with a large field of view (Supplementary Fig. 2) in the supporting information.

The manuscript was modified as follows:

Page 4:

The V_2O_5 nanobelt exhibits unique 1D structure with length of 1 to 5 μm and width of 10 to 100 nm (Supplementary Fig. 1B, D). The lengths of $M_xV_2O_5 \cdot nH_2O$ are between 1 to 10 μm and the widths are between 200 to 450 nm (Fig. 1A, B, Supplementary Fig. 2, 3).

Fig. 1 Synthesis and characterization of $M_xV_2O_5 \cdot nH_2O$ nanobelts. (A) SEM images, (B) TEM images, (C) HRTEM images (the inserts are SAED images) with clear lattice fringes and lattice disorder (D), EDX-mapping (E) of MgVO (a), CaVO (b) and SrVO (c). (F) XRD pattern of MgVO, CaVO, SrVO, V_2O_5 nanobelt and V_2O_5 powder. (G) Crystal structure of α - V_2O_5 (a) and $M_xV_2O_5 \cdot nH_2O$ (b).

The data were added in supporting information:

Supplementary Fig. 2 The TEM images with a large field of view of MgVO (A), CaVO (B) and SrVO (C).

Supplementary Fig. 3 The width distributions of MgVO (A), CaVO (B) and SrVO (C).

4. The key challenges faced by nanozymes are selectivity and specificity. These aspects are currently under active investigation by researchers working in the artificial enzyme area. While the authors assert that their nanozymes exhibit excellent and specific peroxidase-mimicking activity compared to oxidase, catalase, and SOD, previous studies suggest that various vanadium-based systems like nanomaterials and metal-organic frameworks also demonstrate haloperoxidase and glutathione peroxidase activities. Furthermore, there are reports highlighting the efficacy of highly efficient peroxidase nanozymes (<https://www.nature.com/articles/s41467-022-32411-z>).

Response: According to your suggestion, we detected the haloperoxidase-mimicking activity and glutathione peroxidase-mimicking activity of $M_xV_2O_5 \cdot nH_2O$. As shown in Supplementary Fig. 15 and 16, $M_xV_2O_5 \cdot nH_2O$ showed glutathione--mimicking peroxidase activity, and no significant haloperoxidase-mimicking activity was detected.

The manuscript was modified as follows:

Page 9:

In addition, previous studies have shown that some vanadium-based nanozymes also demonstrate haloperoxidase and glutathione peroxidase activities^{49,50}. Therefore, we detected the haloperoxidase-mimicking activity and glutathione peroxidase-mimicking activity of $M_xV_2O_5 \cdot nH_2O$. As shown in Supplementary Fig. 15 and 16, $M_xV_2O_5 \cdot nH_2O$ show glutathione-mimicking peroxidase activity, and no significant haloperoxidase-mimicking activity was detected.

The data were added in supporting information:

Supplementary Fig. 15 Exploration of glutathione peroxidase-mimicking activity of $M_xV_2O_5 \cdot nH_2O$. Time-dependent UV-vis absorption spectra for monitoring the glutathione peroxidase-mimicking catalytic activities of MgVO (A), CaVO (B) and SrVO (C), under the condition of phosphate buffer (pH 7.4) containing 2 mM GSH, 0.4 mM H_2O_2 , and 0.4 mM NADPH at room temperature. Time-dependent UV-vis absorption spectra for monitoring the glutathione peroxidase-mimicking catalytic activities of $M_xV_2O_5 \cdot nH_2O$, under the above condition without GSH (D) or without H_2O_2 (E).

Supplementary Fig. 16 Exploration of haloperoxidase-mimicking activity of $M_xV_2O_5 \cdot nH_2O$. Time-dependent UV-vis absorption spectra for monitoring the glutathione peroxidase-mimicking catalytic activities of MgVO (A), CaVO (B) and SrVO (C), under the condition of NaAc/HAc buffer (pH = 5.0) containing 28 μ M phenol red, 4.4 mM NH_4Br , and 0.42 mM H_2O_2 at room temperature.

5. While most nanozymes exhibit peroxidase-like activity through oxidation by OH radicals, this differs from natural peroxidase enzymes, which utilize ferryl oxo species for oxidation instead of OH radicals.

Response: We are very sorry that we did not make a necessary discussion on the mechanism of POD-like nanozyme in the original text. We have added the mechanism discussion and related references in the revised manuscript. The definition of nanomaterials with POD-like activity usually means that the nanozymes are able to convert H_2O_2 into reactive oxygen species (ROS), which then oxidizes the substrate. Theoretical calculations and some experimental studies have showed that the catalytic reaction paths of POD-like nanozymes can be classified as two types according to the dissociation mode of H_2O_2 adsorbate ($H_2O_2^*$) (Adv. Mater., 2024, 36, 2211151; ACS Nano 2024, 18, 19, 12367). The path 1 is that the dissociated $H_2O_2^*$ generates hydroxyl adsorbate (OH^*) and hydroxyl radical ($\cdot OH$), then generates H_2O and oxidized substrate to complete the cycle (Nat. Nanotechnol., 2007, 2, 577; Angew. Chem. Int. Ed, 2019, 131, 4965; ACS Catal., 2020, 10, 6422; J. Am. Chem. Soc., 2021, 143, 2660; ACS nano, 2022, 16, 4536-4550). Because it is similar to the well-known Fenton reaction, this mechanism is also known as the Fenton-like mechanism and is widely accepted to explain the POD-like activity of many materials. In contrast, path 2 usually does not involve the formation of $\cdot OH$. $H_2O_2^*$ first generates two OH^* , which can directly oxidize the substrate under acidic conditions (ACS Catal., 2020, 10, 12657; J. Am. Chem. Soc., 2021, 143, 8855). Alternatively, OH^* can also be converted to O^* and H_2O^* through a hydrogen transfer reaction, with O^* oxidizing the substrate (Biomaterials, 2015, 48, 37; J. Am. Chem. Soc., 2021, 143, 2660; Nat. Commun., 2022, 13, 4744). This path that adsorbed ROS directly acting on substrates has attracted more and more attention because it is similar to the catalytic mechanism of natural peroxidase (ferryl oxo species). In the revised manuscript, both mechanisms have been described, and corresponding theoretical calculations have also been supplemented and discussed.

The manuscript was modified as follows:

Page 11:

Theoretical calculations and some experimental studies have showed that the catalytic reaction paths of POD-like nanozymes can be classified as two types according to dissociation mode of H_2O_2 adsorbate (H_2O_2^*)^{55,56}. The path 1 is that the dissociated H_2O_2^* generates hydroxyl adsorbate (OH^*) and hydroxyl radical ($\cdot\text{OH}$), then generates H_2O and oxidized substrate to complete the cycle^{31,52,57-59}. Because it is similar to the well-known Fenton reaction, this mechanism is also known as the Fenton-like mechanism and is widely accepted to explain the POD-like activity of many materials. The path 2 usually does not involve the formation of $\cdot\text{OH}$. H_2O_2^* first generates two OH^* , which can directly oxidize the substrate under acidic conditions^{60,61}. Alternatively, OH^* can also be converted to O^* and H_2O^* through a hydrogen transfer reaction, with O^* oxidizing the substrate^{54,58,62,63}. This path that the adsorbed ROS directly acting on substrates has attracted more and more attention because it is similar to the catalytic mechanism of natural peroxidase (ferryl oxo species). In view of the direct detection of $\cdot\text{OH}$ produced under catalysis by ESR spectroscopy and the scavenging verification $\cdot\text{OH}$ by isopropyl alcohol, the possible catalytic mechanism of $\text{M}_x\text{V}_2\text{O}_5 \cdot n\text{H}_2\text{O}$ is proposed following path 1. Therefore, the reaction mechanism was studied from the free energy, charge density difference, and band structure through density functional theory calculation. Subsequently, combined with the changes in the structure and catalytic activity of $\text{M}_x\text{V}_2\text{O}_5 \cdot n\text{H}_2\text{O}$ after M^{2+} intercalation, the synergistic effect of redox-inert M^{2+} with redox-active V is further discussed.

Page 12:

Besides the above mechanism involving releasing $\cdot\text{OH}$, there are many reports on mechanism that the OH^* oxidizes substrates directly under acidic conditions^{60, 61}. In this path, H_2O_2^* first generates two OH^* , and then the reductive substrates (such as TMB) continuously providing reductive hydrogen [H] to the two adsorbed OH^* . Since the presence of the substrate, it is considered to be more similar in energy to the actual catalytic process. Based on this, we calculate the free energy profile under this path. As shown in Supplementary Fig. 24, the dissociation of H_2O_2 is not affected after the substrate is added. Compared with V_2O_5 and $\text{V}_2\text{O}_5 \cdot n\text{H}_2\text{O}$, the energy of H_2O_2 dissociation decreases significantly after Mg/Ca/Sr intercalation. This is consistent with the result in path 1 (Fig. 4B). The reaction of OH^* with TMB-H^+ becomes completely spontaneous due to the formation of H_2O . Therefore, from the perspective of free energy change, $E_{\text{ad}}(\text{OH}^*)$, that is, the stability of OH^* on the catalyst surface, is key to determine the catalytic reaction activity in both path 1 (Fig. 4B) and path 2 (Supplementary Fig. 24).

The data were added in supporting information:

Supplementary Fig. 24. Free energy diagram of the proposed reaction pathway with substrate (TMB) for V₂O₅, V₂O₅·nH₂O and M_xV₂O₅·nH₂O models.

6. The V_{max} and K_M value bars should be presented with error bars. Additionally, it should be noted that there are significant errors in evaluating the kinetic parameters. For instance, the V_{max} for SrVO nanobelts is indicated as 71.7 x 10⁻⁸ M.s⁻¹ (see supporting information Figure 9B(b)); however, this value is notably lower than the plateau region in the plot, which is not possible. Similarly, K_M should ideally reflect the concentration at half of the actual V_{max}, and these values significantly differ in most of the kinetic plots. Authors should carefully examine these aspects as these values are used for comparison with other reported nanozymes.

Response: Thank you very much for your professional reminder.

The K_m and V_{max} were calculated based on the mean values of the reaction rates by three independent steady-state kinetic assays and substrate concentrations. So, the error bars will not be presented in the calculated data. However, both the plots of reaction rates and the double reciprocal plot showed the errors bars caused by the three independent experiments, and detailed raw data about the kinetic parameter have been also provided in the Excel file named “Source Data”.

We have carefully checked the kinetic parameters in the original manuscript and found that some errors in the fitted data, especially for SrVO. In the revised version we have re-fitted all the dynamic parameters and modified the contents related to them. It should be emphasized that the re-fitted data show the relative order of V_{max} corresponding to M_xV₂O₅·nH₂O (M = Mg, Ca, Sr) is same as that in the previous version, which is also consistent with the experimental results.

The manuscript was modified as follows:

Page 9:

As shown in Fig. 3F, when TMB was used as substrate, V_{max} of MgVO, CaVO and SrVO nanobelts are 245, 220, 150 (10⁻⁸ M·s⁻¹), K_m are 0.44, 0.54, 0.27 mM, respectively; When H₂O₂ was used as substrate (Fig. 3G), the V_{max} of MgVO, CaVO

and SrVO nanobelts are 306, 177, 148 ($10^{-8} \text{ M}\cdot\text{s}^{-1}$), and K_m are 7.91, 6.46, 10.32 mM, respectively.

Fig. 3 (D) Time-dependent absorbance of oxTMB (a), reaction rate (b) with different concentrations of TMB and fixed concentrations of MgVO/H₂O₂, and the corresponding double reciprocal (Lineweaver-Burk) plots (inset). (E) Time-dependent absorbance of oxTMB (a), reaction rate (b) with different concentrations of H₂O₂ and fixed concentrations of MgVO/TMB, and the corresponding double reciprocal (Lineweaver-Burk) plots (inset). V_{max} and K_m values of $M_xV_2O_5 \cdot nH_2O/V_2O_5$ nanobelts towards to TMB (F) and (G) H₂O₂.

The supporting information was modified as follows:

Supplementary Fig. 17 (A) Time-dependent absorbance of oxTMB (a), reaction rate (b) with different concentrations of TMB and fixed concentrations of CaVO/H₂O₂, and the corresponding double reciprocal (Lineweaver-Burk) plots (c). (B) Time-dependent absorbance of oxTMB (a), reaction rate (b) with different concentrations of H₂O₂ and fixed concentrations of CaVO/TMB, and the corresponding double reciprocal (Lineweaver-Burk) plots (c). Error bars represent \pm SD ($n = 3$)

Supplementary Fig. 18 (A) Time-dependent absorbance of oxTMB (a), reaction rate (b) with different concentrations of TMB and fixed concentrations of SrVO/ H_2O_2 , and the corresponding double reciprocal (Lineweaver-Burk) plots (c). (B) Time-dependent absorbance of oxTMB (a), reaction rate (b) with different concentrations of H_2O_2 and fixed concentrations of SrVO/TMB, and the corresponding double reciprocal (Lineweaver-Burk) plots (c). Error bars represent $\pm SD$ ($n = 3$)

Supplementary Fig. 19 (A) Time-dependent absorbance of oxTMB (a), reaction rate (b) with different concentrations of TMB and fixed concentrations of V_2O_5 nanobelts/ H_2O_2 , and the corresponding double reciprocal (Lineweaver-Burk) plots (c). (B) Time-dependent absorbance of oxTMB (a), reaction rate (b) with different concentrations of H_2O_2 and fixed concentrations of V_2O_5 nanobelts/TMB, and the corresponding double reciprocal (Lineweaver-Burk) plots (c). Error bars represent $\pm SD$ ($n = 3$)

7. Authors have compared the activities of other peroxidase mimics, but the basis of comparison remains unclear. Considering the differences in morphologies, sizes, and structures among these systems, it becomes challenging to make direct comparisons.

Utilizing the quantity of materials while maintaining a constant surface area for all samples may be more relevant in this case.

Response: When comparing the catalytic activity of nanozymes, the definition of nanozyme unit is crucial for different comparison methods. Among them, considering each atom/molecule in the nanoparticle as a nanozyme unit and each surface atom in the nanoparticle as a nanozyme unit are two common ways in the literatures (Adv. Mater., 2024, 36, 2211041). In general, considering each molecule in the nanoparticle ($M_xV_2O_5 \cdot nH_2O$, in this work) as a nanozyme unit is thought to result in underestimation of nanozyme activity. The method using surface atom as nanozyme unit (surface V, as your suggestion) is considered to better reflect the intrinsic catalytic activity of nanozymes under the premise that the activity depends only on the surface active sites. However, the determination of the surface atomic number first requires the surface area of the material, which can be obtained from the geometry calculations of nanomaterials or techniques such as Brunauer-Emmett Teller (BET) adsorption isotherms, and then the surface atomic number can be obtained according to the surface area and atomic arrangement of the crystal plane (ACS nano, 2021, 15, 15645). This way is usually suitable for nanoparticles with good monodispersity and uniform size. In this work, due to the large geometric heterogeneity of $M_xV_2O_5 \cdot nH_2O$, there may be significant errors in the measurement of surface atoms. In addition, it has been reported that the activity of nanozyme is not only determined by the surface atoms, the internal atoms may participate in catalytic regulation through electron transfer (Nat. Commun., 2022, 13, 5365). More importantly, in this work, since redox-inert M^{2+} as Lewis acid participated in the POD-like catalysis of $M_xV_2O_5 \cdot nH_2O$, each $M_xV_2O_5 \cdot nH_2O$ molecule in the particle as a nanozyme unit can better reflect the synergistic effect of M^{2+} and V. Therefore, the activity comparison method adopted in this paper may be more suitable for our synthesized $M_xV_2O_5 \cdot nH_2O$. We have added relevant literature to the revised manuscript and accurately described the concentration of the nanozyme unit we used in the experimental method.

The manuscript was modified as follows:

Page 11:

In order to systematically evaluate its catalytic performance, we compared the V_{max} and TON values (the maximum number of conversing substrates via the mole concentration of metal in the whole nanomaterials⁵¹) of $M_xV_2O_5 \cdot nH_2O$ with currently reported classical peroxidase mimics,

Page 19:

$TON = V_{max}/[E]$, where $[E]$ is the molar concentration of MgVO/CaVO/SrVO.

8. In Figure 4a, hydrogen peroxide should be adsorbed and not absorbed. This correction should be applied throughout the figure and text. Moreover, in step 2 of this figure, the term "hemlysis" used by the authors requires clarification. After step 3, the OH radical is depicted near the surface of the material, but its reaction with a proton (which should be proton adsorption) results in a water molecule. If this is the case, the

generated water molecule would lack one electron, which is not feasible. Authors should carefully review the mechanism.

Response: We are very sorry for the spelling errors. The "absorption" and "hemlysis" in Fig. 4 A and C have been revised to "adsorption" and "homolysis", respectively. The corresponding words in the manuscript have also been checked and modified.

For the actual catalysis, the electrons obtained by $\cdot\text{OH}$ generally come from the substrate or the catalyst (electron transfer). However, in the free energy profile in this paper, due to the use of calculation models that do not consider the substrate, the electrons directly derived from the substrate or the catalyst are not reflected in the mechanism diagram. Besides, in the free energy calculation using density functional theory, only the free energy of specific states such as H_2O_2 adsorption or OH adsorption was calculated, and no electron transport was involved. In addition, we have added the free energy profile considering TMB substrate (Supplementary Fig. 24), where the electrons are concerned

As for the mechanism, we have added relevant content in the revised manuscript and introduced it in Response 5. Based on your suggestion and our further analysis on the reaction mechanism, the Fig. 4 A has been modified as followed:

Fig. 4 (A) The possible reaction pathway on $\text{M}_x\text{V}_2\text{O}_5 \cdot n\text{H}_2\text{O}$, taking MgVO for example. (Purple: V atom; Red: O atom; White: H atom; Yellow: Mg atom).

The data were added in supporting information:

Supplementary Fig. 24 Free energy diagram of the proposed reaction pathway with substrate (TMB) for V_2O_5 , $\text{V}_2\text{O}_5 \cdot n\text{H}_2\text{O}$ and $\text{M}_x\text{V}_2\text{O}_5 \cdot n\text{H}_2\text{O}$ models.

9. Authors should also furnish evidence regarding the oxidation states of metal centers during catalysis. Is there any change in the oxidation state when TMB interacts with the catalyst and H₂O₂?

Response: In order to evaluate the oxidation state change of V in M_xV₂O₅·nH₂O during catalysis, the M_xV₂O₅·nH₂O were dropped on glassy carbon electrodes (GCE) and the cyclic voltammetry (CV) curves were measured in 3 M KCl solution with H₂O₂ and TMB as electrolyte (Nat. Commun., 2021, 12, 3375; ACS nano, 2019, 13, 1870). At the same time, CV curves of M_xV₂O₅·nH₂O modified GCE electrodes were measured in 3 M KCl solution as controls. As shown in Supplementary Fig. 12 A, each CV curve of the M_xV₂O₅·nH₂O electrode presents two pairs of redox peaks, indicating the conversion of V between different oxidation states (Angew. Chem. Int. Ed. 2005, 44, 4391). Compared with the 3 M KCl solution as electrolyte, the presence of H₂O₂ and TMB in electrolyte significantly reduces the redox current. These results show that the electron transfer between different oxidation states of V participates in the catalytic reaction, which competes with the electrode reaction, leading to reduced current (Supplementary Fig. 12, 13).

In order to verify the effect of catalytic reaction on the oxidation states of V, XPS analysis were performed on M_xV₂O₅·nH₂O before and after the reaction. The results in Supplementary Fig. 20 show there is little change in the ratio of V with different oxidation states before and after the catalytic reaction.

The manuscript was modified as follows:

Page 9:

Meanwhile, cyclic voltammetry was used to verify the catalytic activity of M_xV₂O₅·nH₂O on the substrate^{27,48}. As shown in Supplementary Fig. 12, 13, when H₂O₂ and TMB are added to the electrolyte, the currents corresponding to the redox of V with different valence states are significantly reduced, showing obvious catalytic responses.

The XPS analysis of M_xV₂O₅·nH₂O was performed before and after the reaction. It can be seen from Supplementary Fig. 20 that M_xV₂O₅·nH₂O remain relatively stable under catalytic conditions.

The data were added in supporting information:

Supplementary Fig. 12 (A) The CV curves of bare GCE and $M_xV_2O_5 \cdot nH_2O$ modified GCE in 3 M KCl, scan rate: 50 mV/s. The CV curves of MgVO (B), CaVO (C) and SrVO (D) modified GCE in 3 M KCl without or with different concentrations of H₂O₂ (0.1- 0.5 mM) and fixed concentrations of TMB (0.2 mM), scan rate: 50 mV/s.

Supplementary Fig. 13 Time-dependent CV curves of MgVO (A), CaVO (B) and SrVO (C) modified GCE in 3 M KCl (scan rate: 50 mV/s) showed that the currents do not change significantly, demonstrating the electrochemical stability of the $M_xV_2O_5 \cdot nH_2O$.

Supplementary Fig. 20 XPS spectra of V 2P of MgVO (A), CaVO (B), and SrVO (C) before and after enzymatic reaction.

10. In line 398, Authors say “the proposed $M_xV_2O_5 \cdot nH_2O$ were extremely similar in structure and function to natural peroxidase”. On what basis are they extremely similar in structure? The mechanism is also quite distinct from natural enzyme (please see comment no. 5).

Response: We are sorry for our imprecise description. In this paper, we emphasize that redox-active V and redox-inert M^{2+} in $M_xV_2O_5 \cdot nH_2O$ have good similarities with redox-active Fe^{3+} and redox-inert Ca^{2+} in natural horseradish peroxidase (HRP), in terms of structural and functional synergism of the two metals. For the structure, the HRP contains high-spin Fe^{3+} , located in protoporphyrin IX in coordination with the proximal histidine ligand. At the same time, there is also redox-inert Ca^{2+} in HRP-C. The Ca^{2+} are on both sides of the protoporphyrin plane with a content of 2 mol of Ca^{2+} /mol of enzyme. Although Fe^{3+} and Ca^{2+} are not chemically bonded, Ca^{2+} can regulate the electronic structure of heme iron. In $M_xV_2O_5 \cdot nH_2O$, redox-inert M^{2+} exist on both sides of the redox-active V as intercalation ions. At the same time, M^{2+} and V are connected by non-covalent bond, but M^{2+} can affect the electron density of V by ion polarization. So, the $M_xV_2O_5 \cdot nH_2O$ is similar to HRP in geometric and electronic structure. For the catalytic function, the introduction of M^{2+} effectively enhanced the POD-mimicking activity of $M_xV_2O_5 \cdot nH_2O$, partial removal of M^{2+} leading to significant reduction of catalytic activity, which is also extremely similar to that, removal of Ca^{2+} resulted in obvious decrease in the activity of the HRP-C (Biochem. Biophys. Res. Commun. 1978, 80, 1039; J. Biol. Chem. 1990, 265, 13335). The description of the structure had been presented in Scheme 1a. The function of M^{2+} as Lewis acid had been experimentally validated. In order to make the expression more accurate, we have changed this sentence in the revised manuscript.

The manuscript was modified as follows:

Page 14:

According to the reported literatures, the redox-inert Ca^{2+} as cofactors in natural peroxidase with Lewis acidity play a key role in the catalytic performance, which is mainly manifested in the loss of Ca^{2+} leading to significant reduction of catalytic activity^{16,17}. The structural similarities between $M_xV_2O_5 \cdot nH_2O$ and natural peroxidase encouraged us to further investigate whether the redox-inert M^{2+} could also act as cofactors like Ca^{2+} of natural peroxidase to affect the peroxidase-mimicking activity of $M_xV_2O_5 \cdot nH_2O$. Thus, $M_xV_2O_5 \cdot nH_2O$ were placed in heated alkaline liquid to remove the inserted M^{2+} to achieve deintercalation. The results of ICP-MS showed that the molar ratios of Mg/V, Ca/V, and Sr/V in $M_xV_2O_5 \cdot nH_2O$ decreased to 0.08/1, 0.13/1, and 0.14/1 respectively (Supplementary Fig. 26, Supplementary Table 4), confirming partial removal of M^{2+} in $M_xV_2O_5 \cdot nH_2O$. The catalytic capacity of $M_xV_2O_5 \cdot nH_2O$ for TMB oxidation before and after de-intercalation were then compared. From the result shown in Fig. 4G, the catalytic performance of $M_xV_2O_5 \cdot nH_2O$ after deintercalation show obvious decrement owing to the decreased concentration of M^{2+} . The SEM and XRD results show that the morphology and crystal structure of $M_xV_2O_5 \cdot nH_2O$ after partial deintercalation have no obvious changes (Supplementary Fig. 27, 28). The above results showed that the redox-inert M^{2+} in $M_xV_2O_5 \cdot nH_2O$ play similar roles to the Ca^{2+}

in natural peroxidase in regulating the catalytic reaction. Benefiting from these, the synergism of redox-active V and redox-inert M^{2+} in $M_xV_2O_5 \cdot nH_2O$ plays a key role in regulating the structure and function of peroxidase mimics, which is extremely similar to the relationship between redox site and Lewis acid in natural peroxidase.

11. The name of the bacteria should always be conventionally written in italics. For example, *Staphylococcus aureus*. Please correct such mistakes.

Response: Thank you for your reminding. We have corrected such mistakes.

12. Authors have only used two bacteria in this study, but they claim that their nanozymes have broad-spectrum activity. Evaluation should be conducted using more strains to substantiate this claim.

Response: Thank you for your advice. We added the antibacterial test for gram-positive *Bacillus subtilis* (*B. subtilis*) and gram-negative *Pseudomonas aeruginosa* (*P. aeruginosa*) by standard spread plate method. As shown in Supplementary Fig. 29, the $M_xV_2O_5 \cdot nH_2O$ show significant antibacterial ability against *B. subtilis* and *P. aeruginosa* in the presence of low concentration H_2O_2 .

The manuscript was modified as follows:

Page 15:

To further verify the broad-spectrum antibacterial ability of $M_xV_2O_5 \cdot nH_2O$, the antibacterial test for gram-positive *Bacillus subtilis* (*B. subtilis*) and gram-negative *Pseudomonas aeruginosa* (*P. aeruginosa*) by standard spread plate method were added. As shown in Supplementary Fig. 29, $M_xV_2O_5 \cdot nH_2O$ show significant antibacterial ability against *B. subtilis* and *P. aeruginosa* in the presence of low concentration H_2O_2 .

The data were added in supporting information:

Supplementary Fig. 29. Photographs of bacterial colonies against *P. aeruginosa* (A) and *B. subtilis* (C) with different treatments. Relative survival rate of *P. aeruginosa* (B) and *B. subtilis* (D) upon different treatments determined by spread plate method. (a) PBS, (b) H_2O_2 , (c) V_2O_5 powder, (d) V_2O_5 nanobelt, (e) MgVO, (f) CaVO, (g) SrVO, (h) V_2O_5 powder + H_2O_2 , (i) V_2O_5 nanobelt + H_2O_2 , (j) MgVO + H_2O_2 , (k) CaVO + H_2O_2 , (l) SrVO + H_2O_2 . Error bars represent \pm SD (n = 3). Data were analyzed by one-way ANOVA with Turkey's multiple comparisons test, ns represents no statistical difference, *P<0.01. **P<0.001. ***P<0.0001. ****P<0.0001.

13. I see that the nanozymes demonstrate antibacterial activities and not bacteriostatic activities. Kindly correct it in line 429.

Response: Thank you very much for your reminding. We have made correction in the text and checked other similar words.

The manuscript was modified as follows:

Page 15:

As shown in Fig. 5E, 5F, compared with other control groups, the red color of PI in the presence of both H_2O_2 and $M_xV_2O_5 \cdot nH_2O$ was enhanced, indicating a significant antibacterial effect.

14. In Fig 6B, the dimensions of the wound (Day 0) in lanes 1, 2, and 3 appear to be different from the others. How can these be correctly compared in the study?

Response: In infected wound healing experiments, although the wound creation followed same criteria, the errors were objectively present. We have shown the wound area statistics of multiple mice in different treatment groups in Supplementary Fig. 34, and the errors of the wound area are within a reasonable range. Because of the influence of shooting angle and mouse fixation, the dimensions of some wounds may look different in the photos. The antibacterial performance of the materials can be verified by the wound healing of the same mouse over a certain period of time. After applying the materials, all mice showed ideal wound healing.

The manuscript was modified as follows:

Page 15, 17:

A round whole cortical wound with a diameter about 5.5 mm was formed on the back of each mouse (Supplementary Fig. 34)

The data were added in supporting information:

Supplementary Fig. 34. The diameters of wounds at day 0 measured by vernier caliper. ns represents no statistical difference. Error bars represent $\pm SD$ ($n = 3$)

15. In Supplementary Figure 18, no belt-like structures are visible in images B, C, and

D. Considering that these belts have lengths of 1-4 μm , they are not visible at the scale size of 5 and 10 μm in the provided SEM images.

Response: According to your suggestion, we re-tested the composite hydrogel by SEM. Since the GelTA is the main component of the composite hydrogel, $\text{M}_x\text{V}_2\text{O}_5 \cdot n\text{H}_2\text{O}$ are only low content additives, it is difficult to directly observe the $\text{M}_x\text{V}_2\text{O}_5 \cdot n\text{H}_2\text{O}$ in the composite hydrogel. In the revised support information, we replaced the previous SEM images with the SEM images containing EDX elemental quantification. The new data can show the elemental composition of the $\text{M}_x\text{V}_2\text{O}_5 \cdot n\text{H}_2\text{O}$ functionalized composite hydrogel, as well as more typical porous structures.

The manuscript was modified as follows:

Page 15:

SEM images showed that freeze-dried GelTA has abundant porous structure with high interconnectivity. $\text{M}_x\text{V}_2\text{O}_5 \cdot n\text{H}_2\text{O}$ functionalized GelTA (Mg-Gel-TA/ Ca-Gel-TA/ Sr-Gel-TA) also displayed similar porous structure, which were beneficial for exchange of substance (Supplementary Fig. 32). EDX elemental quantification shows the existence of $\text{M}_x\text{V}_2\text{O}_5 \cdot n\text{H}_2\text{O}$ in the composite hydrogel.

The supporting information was modified as follows:

Supplementary Fig. 32 The SEM images of GelTA (A) and SEM images with EDX elemental quantification (insets) of Mg-GelTA (B), Ca-GelTA (C), Sr-GelTA (D).

16. Authors have compared their nanozymes' activities with HRP and Fe_3O_4 reported by Gao et al. However, this comparison may not be appropriate as HRP and Fe_3O_4 have been evaluated under different reaction conditions. Therefore, it is improper to conclude that “the peroxidase-mimicking activity of $\text{M}_x\text{V}_2\text{O}_5 \cdot n\text{H}_2\text{O}$ was more than one order of magnitude higher”.

Response: We are very sorry for the inaccurate statement. The V_{\max} is only a reference index of enzyme activity and can not be used to directly compare the catalytical activity of materials. In the revised manuscript, we have revised the above sentence.

The manuscript was modified as follows:

Page 9-11:

Comparing with K_m and V_{\max} of the natural horseradish peroxidase (HRP) and classical Fe_3O_4 (Supplementary Table 2), the K_m of $M_xV_2O_5 \cdot nH_2O$ for the two substrates are similar to that of HRP and Fe_3O_4 , but the V_{\max} of $M_xV_2O_5 \cdot nH_2O$ are more than one order of magnitude higher than that of HRP and Fe_3O_4 under their respective optimal catalytic conditions.

17. There are grammatical errors in multiple places.

Response: Thank you very much for your reminding. We have re-checked the full text and made modifications to many non-standard sentences.

Reviewer #2 (Remarks to the Author):

Lewis acids such as Ca^{2+} play multifaceted roles in biological enzymes, especially in the catalytic action of peroxidases, including stabilizing the active site structure, promoting substrate binding, regulating the catalytic process, stabilizing the enzyme-substrate complex, and adjusting enzyme activity. This provides a good perspective for the rational design of nanozymes, but current research on nanozyme designs that consider the auxiliary role of Lewis acids is limited, so the creativity of this paper is innovative. However, there are still some issues that need improvement, and I suggest considering publication in Nature Communications after major revision. Here are my suggestions:

1. Please clarify the long-term stability of the material under acidic conditions. Does the material degrade?

Response: Thanks very much for your reminding. In the revised manuscript, we have added the long-term stability verification of $M_xV_2O_5 \cdot nH_2O$ at $pH=5$. The added XRD and SEM data (Supplementary Fig. 9, 10) showed that $M_xV_2O_5 \cdot nH_2O$ maintained stable crystal structures and morphologies within 15 days.

The manuscript was modified as follows:

Page 8:

Experiments also show that $M_xV_2O_5 \cdot nH_2O$ nanobelts exhibit optimal peroxidase-mimicking catalytic activity under weak acidic environment ($pH = 5$) and near room temperature ($10-80\text{ }^\circ\text{C}$) (Supplementary Fig. 8). The $M_xV_2O_5 \cdot nH_2O$ nanobelts maintain stable crystal structures and morphologies within 15 days at $pH = 5$ (Supplementary Fig. 9, 10).

The data were added in supporting information:

Supplementary Fig. 9 The XRD patterns of MgVO (A), CaVO (B) and SrVO (C) under HAc-NaAC buffer (0.2 M, pH = 5.0) within 15 days.

Supplementary Fig. 10 The SEM images of MgVO (A), CaVO (B) and SrVO (C) under HAc-NaAC buffer (0.2 M, pH = 5.0) within 15 days.

2. The theoretical calculation part of the mechanism does not consider the substrate, such as TMB. The adsorbed two HO^* are directly desorbed, and one combines with H^+ in the solution to get an electron from the material to form water and desorb. I don't think this mechanism is very reasonable because: a. The relative activity of the nanozyme is related to the substrate, so the substrate should be considered. b. The calculation results of this paper show that the adsorption of HO^* is an endothermic process, so the free $\bullet\text{OH}$ adsorption on the surface is a spontaneous exothermic process. Therefore, compared to the mechanism of releasing free radicals, I think the process of continuously providing reductive hydrogen [H] to the two adsorbed HO^* by reductive substrates, such as TMB, reported in the previous literature (Advanced Materials 2023, 2211151.) is more reasonable.

Response: Thank you very much for your professional suggestion. We have referred the relevant literatures to add the free energy profile considering substrate in the reaction mechanism and discuss it (Adv. Mater., 2024, 36, 2211151; ACS Catal., 2020, 10, 12657). In this path, H_2O_2^* first generates two OH^* , and then the reductive substrates (such as TMB) continuously providing reductive hydrogen [H] to the two adsorbed HO^* . Based on this, we obtained the free energy profile. As shown in Supplementary Fig. 24, the dissociation of H_2O_2 is not affected after the substrate is added. Compared with V_2O_5 and $\text{V}_2\text{O}_5 \cdot n\text{H}_2\text{O}$, the energy of H_2O_2 dissociation decreases significantly after Mg/Ca/Sr intercalation. This is consistent with the result in path 1 (Fig. 4B). The reaction of OH^* with TMB-H^+ becomes completely spontaneous due to the formation of H_2O . Therefore, from the perspective of free energy change, $E_{\text{ad}}(\text{OH}^*)$, that is, the stability of OH^* on the catalyst surface, is key

to determine the catalytic reaction activity in both path 1 (Fig. 4B) and path 2 (Supplementary Fig. 24).

At the same time, we still keep the free energy calculation without adding substrate according to the research goal of this work. In view of the complexity of the actual reaction, it is difficult for theoretical calculation to fully simulate the actual situation in the theoretical calculation. For POD-like nanozyme and natural POD, it has the ability to catalyze H₂O₂-dependent oxidation of a variety of substrates. Therefore, we believe that, under the condition without specific substrate, studying the intrinsic catalytic decomposition ability of nanozyme to H₂O₂ can also provide a valuable perspective for enzyme-like catalytic reaction mechanism. In this work, we focus on the biomimetic synergistic effect of redox site and Lewis acid, and try to propose a new strategy for constructing efficient artificial enzymes. The calculation ignoring substrate may have more advantages in analyzing material characteristics. In addition, we also revised the calculation models according to your comment 5 and 6, focusing on calculating the change of free energy before and after ion intercalation and discussing its effect on the catalytic activity of enzymes.

The manuscript was modified as follows:

Page 12:

Besides the above mechanism involving releasing ·OH, there are many reports on mechanism that the OH* oxidizes substrates directly under acidic conditions^{60,61}. In this path, H₂O₂* first generates two OH*, and then the reductive substrates (such as TMB) continuously providing reductive hydrogen [H] to the two adsorbed OH*. Since the presence of the substrate, it is considered to be more similar in energy to the actual catalytic process. Based on this, we calculate the free energy profile under this path. As shown in Supplementary Fig. 24, the dissociation of H₂O₂ is not affected after the substrate is added. Compared with V₂O₅ and V₂O₅·nH₂O, the energy of H₂O₂ dissociation decreases significantly after Mg/Ca/Sr intercalation. This is consistent with the result in path 1 (Fig. 4B). The reaction of OH* with TMB-H⁺ becomes completely spontaneous due to the formation of H₂O. Therefore, from the perspective of free energy change, E_{ad}(OH*), that is, the stability of OH* on the catalyst surface, is key to determine the catalytic reaction activity in both path 1 (Fig. 4B) and path 2 (Supplementary Fig. 24).

The data were added in supporting information:

Supplementary Fig. 24. Free energy diagram of the proposed reaction pathway with substrate (TMB) for V_2O_5 , $V_2O_5 \cdot nH_2O$ and $M_xV_2O_5 \cdot nH_2O$ models.

3. The adsorption energy of H_2O_2 for noble metals, metal oxides, carbon-based materials, etc., is roughly between 0.0 and -1.0. However, the calculated adsorption energy of H_2O_2 in this paper is very strong, reaching -4.0 to -5.0 eV, which has obviously exceeded the strength of chemical adsorption, indicating that new chemical bonds have been formed. Please confirm the accuracy of the calculation, explain the reasons, or are there any literature reports of similar results?

Response: In the previous calculation, we built calculation models that included lattice oxygen vacancies. The simultaneous existence of lattice oxygen vacancy and surface unsaturated coordination may cause the adsorption models unstable. H_2O_2 will fill the oxygen vacancy after adsorption on the surface, resulting in low adsorption energy. In the revised draft, we remove the additional lattice oxygen vacancies to optimize the models. The adsorption energy of H_2O_2 for each model is within the reasonable range of 0.0 to -1.0. In addition, according to your comment 5 and 6, we modified the relevant crystal models and added a new control model to obtain a new free energy profile. The relevant content has been added in the main text.

The manuscript was modified as follows:

Fig. 4. (B) Free energy diagram of the proposed reaction pathway for V_2O_5 , $V_2O_5 \cdot nH_2O$ and $M_xV_2O_5 \cdot nH_2O$ models.

The data were added in supporting information:

Supplementary Fig. 23. Top and side views of five crystal models including perfect V_2O_5 , $V_2O_5 \cdot nH_2O$, MgVO, CaVO, and SrVO.

4. There are already many theoretical reports on the mechanism of POD activity of nanozymes, but the authors rarely cite and discuss them.

Response: Thank you very much for your suggestion, we have added relevant content and literature in the revised manuscript. The definition of nanomaterials with POD-like activity usually means that the nanozymes are able to convert H_2O_2 into reactive oxygen species (ROS), which then oxidizes the substrate. Theoretical calculations and some experimental studies have showed that the catalytic reaction paths of POD-like nanozymes can be classified as two types according to dissociation mode of H_2O_2 adsorbate ($H_2O_2^*$) (Adv. Mater., 2024, 36, 211151; ACS Nano 2024, 18, 12367). The path 1 is that the dissociated $H_2O_2^*$ generates hydroxyl adsorbate (OH^*) and hydroxyl radical ($\cdot OH$), and generates H_2O and oxidized substrate to complete the cycle (Nat. Nanotechnol., 2007, 2, 577; Angew. Chem. Int. Ed, 2019, 131, 4965; ACS Catal., 2020, 10, 6422; J. Am. Chem. Soc., 2021, 143, 2660; ACS nano, 2022, 16, 4536). Because it is similar to the well-known Fenton reaction, this mechanism is also known as the Fenton-like mechanism and is widely accepted to explain the POD-like activity of many materials. Path 2 usually does not involve the formation of $\cdot OH$. $H_2O_2^*$ first generates two OH^* , which can directly oxidize the substrate under acidic conditions (ACS Catal.,

2020, 10, 12657; J. Am. Chem. Soc., 2021, 143, 8855). Alternatively, OH* can also be converted to O* and H₂O* through a hydrogen transfer reaction, with O* oxidizing the substrate (Biomaterials, 2015, 48, 37; J. Am. Chem. Soc., 2021, 143, 2660; Nat. Commun., 2022, 13, 4744). This path that the adsorbed ROS directly acting on substrates has attracted more and more attention because it is similar to the catalytic mechanism of natural peroxidase (ferryl oxo species).

The manuscript was modified as follows:

Page 11:

Theoretical calculations and some experimental studies have showed that the catalytic reaction paths of POD-like nanozymes can be classified as two types according to dissociation mode of H₂O₂ adsorbate (H₂O₂*)^{55,56}. The path 1 is that the dissociated H₂O₂* generates hydroxyl adsorbate (OH*) and hydroxyl radical (·OH), then generates H₂O and oxidized substrate to complete the cycle^{31,52,57-59}. Because it is similar to the well-known Fenton reaction, this mechanism is also known as the Fenton-like mechanism and is widely accepted to explain the POD-like activity of many materials. The path 2 usually does not involve the formation of ·OH. H₂O₂* first generates two OH*, which can directly oxidize the substrate under acidic conditions^{60,61}. Alternatively, OH* can also be converted to O* and H₂O* through a hydrogen transfer reaction, with O* oxidizing the substrate^{54,58,62,63}. This path that the adsorbed ROS directly acting on substrates has attracted more and more attention because it is similar to the catalytic mechanism of natural peroxidase (ferryl oxo species). In view of the direct detection of ·OH produced under catalysis by ESR spectroscopy and the scavenging verification ·OH by isopropyl alcohol, the possible catalytic mechanism of M_xV₂O₅·nH₂O is proposed following path 1. Therefore, the reaction mechanism was studied from the free energy, charge density difference, and band structure through density functional theory calculation. Subsequently, combined with the changes in the structure and catalytic activity of M_xV₂O₅·nH₂O after M²⁺ intercalation, the synergistic effect of redox-inert M²⁺ with redox-active V is further discussed.

5. Why didn't the authors consider the effect of surface Ca²⁺, instead of interlayer Ca²⁺, on enzyme activity?

Response: Due to the large difference in valence state and outer electronic structure between alkali earth metal M and transition metal V, M²⁺ is theoretically not inclined to be incorporated into the host lattice of V₂O₅ in the form of covalent bonds. The M²⁺ in M_xV₂O₅·nH₂O is an intercalating doping based on non-covalent bonding, which is verified by XRD and XPS. This weak interaction is relatively stable between layers, but unstable on the surface. In addition, M²⁺ is considered to have no catalytic ability. The redox-inert M²⁺ should improve the POD-like catalytic activity by regulating electronic structure of V. Therefore, interlayer M²⁺, as the main existing form of M²⁺ in the layered materials, is usually not considered to directly contact substrates on material surface, but rather achieves synergistic effect with surface V.

6. Using Lewis acid assistance to improve enzyme activity and perform structure-

activity relationship analysis is an interesting topic, but the authors did not clearly explain the actual impact of Lewis acid on VO materials, simply explaining it as the polarization effect of ionic potential on VO. The effect from electronic effects is reasonable, but it should be thoroughly explained. I suggest that the authors calculate the band structure of the models and pay attention to the frontier energy levels, and discuss the activity in combination with the redox potential, because the essence of the substrate [H] snatching is a redox reaction. Therefore, the impact of Lewis acid on activity can be discussed from the perspective of the influence of electronic effects on the redox potential, which can be done by comparing the differential charge density analysis before and after addition, rather than simply analyzing the electron transfer of OH*, because there is no significant difference in the results of figure 4Fc, d, e as it stands.

Response: According to your suggestion, we calculated the frontier molecular orbitals (FMO) of the materials before and after intercalation. Then, the changes of POD-like activity were analyzed by combining the band structures and the redox potential of POD-like catalytic reaction (Nat. Commun., 2021, 12, 6866; ACS Appl. Mater. Interfaces, 2014, 6, 1959). The experimental results showed that the FMO energy of the material moved towards negative potential after Lewis acid intercalation, indicating that the reduction ability of $M_xV_2O_5 \cdot nH_2O$ was significantly enhanced. At the same time, the valence orbital maximums of $M_xV_2O_5 \cdot nH_2O$ are close to or lower than the reduction potential of H_2O_2 (Fig. 5 F). According to relevant literature (Nat. Commun., 2021, 12, 6866; ACS Appl. Mater. Interfaces, 2014, 6, 1959) and Nernst equation, this band structure is conducive to electron transfer in the catalytic reaction to achieve enzyme-like catalysis. We also analyzed the intrinsic band charge of the material before and after the Lewis acid intercalation. The experimental results show that the polarization of Lewis acid can reduce the electron density around V, make V more inclined to form V-OH, and promote the H_2O_2 homolysis (Supplementary Fig. 25).

The manuscript was modified as follows

Page 13:

In order to further understand the effect of Lewis acid intercalation on enzyme-mimicking catalytic activity, the band structure before and after intercalation were calculated and discussed combining with the POD-like catalytic redox potential^{66,67}. The typical reaction catalyzed by peroxidase nanozyme is as follows: $H_2O_2 + 2TMB + 2H^+ = 2H_2O + 2TMB^+$, where TMB^+ is the oxidation state of TMB. The reaction can be divided into two half reactions as follows:

The reduction potential of TMB^+/TMB (φ_1) used is about 1.13V, referring to a well-established value in the literature, and the standard reduction potential of H_2O_2/H_2O (φ_2) is 1.776V^{68,69}. As can be seen from Fig. 4F, the frontier molecular orbitals (FMO) of V_2O_5 before intercalation, including the valence band maximum (VBM) and the conduction band minimum (CBM), display more positive energy than φ_2 , indicating that electrons can be transferred from TMB to V_2O_5 , but cannot be transferred from V_2O_5 to H_2O_2 to complete the catalytic reaction. After Lewis acid M^{2+} intercalation, the

energies of the FMO of $M_xV_2O_5 \cdot nH_2O$ move significantly toward the negative direction, indicating that the reduction abilities of $M_xV_2O_5 \cdot nH_2O$ are significantly enhanced. Among them, the energies of VBM is close to or less negative than ϕ_2 , indicating that electrons can be passed from TMB to V_2O_5 , and then to H_2O_2 to complete the catalysis. The H_2O intercalated $V_2O_5 \cdot nH_2O$ shows a more negative FMO energy, and electrons cannot transfer from TMB to V_2O_5 .

Fig. 5 (F) Calculated electronic density of states with the energies of FMOs marked

Then, the influence of Lewis acid M^{2+} on the intrinsic electronic structure of redox-active V is further analyzed by calculating the average bader charge of V before and after intercalation. As shown in Supplementary Fig.25, the bader charge around V in V_2O_5 before intercalation is 2.1670 |e|. After H_2O intercalation, the bader charge around V is 2.1730 |e|, and no significant change occurs. After M^{2+} intercalation, the average bader charge around V decreases to 2.1379, 2.1264 and 2.1242 |e|, respectively. Since M^{2+} acts directly on the coordinated O of V through electrostatic attraction, the polarization of M^{2+} on the V-O bond results in the decrease in electron density around V. In general, V with low electron density will be more inclined to produce stable V-OH*. Thus, the Lewis acid M^{2+} promotes the $H_2O_2^*$ homolysis by adjusting the electronic structure of V.

The data were added in supporting information:

Supplementary Fig. 25 The bader charge and differential charge analysis of V_2O_5 (A), MgVO (B), CaVO (C), SrVO (D) and $V_2O_5 \cdot nH_2O$ (E). (cyan and yellow represent charge depletion and accumulation, respectively)

7. Line 324-325, the authors did not calculate the transition state, so I don't understand the term "barrier of decomposition", and why the barriers are all negative, which seems inconsistent with the results in the figure.

Response: We are very sorry for the mistakes in the use of professional terms, thank you for your professional reminder. In the revised manuscript, we have recalculated the free energy and corrected the relevant expression in combination with your other suggestions.

8. How did the authors calculate the charged system, by adding a background charge? Please supplement the explanation in the calculation method.

Response: Thank you very much for your reminding. In this work, H^+ and $TMBH^+$ are involved in the calculation of free energy. In fact, the first-principle calculations based on DFT are difficult to deal with the charged system, and the energy obtained by directly adding a background charge correction may be inaccurate. In addition, the VASP official manual clearly states that the energy obtained by adding background charges cannot be trusted. Based on this, $TMBH^+$ is calculated as TMBH radical. However, the free energy of H^+ is not directly calculated, but obtained by $H^+ + e^- \rightarrow 0.5 H_2$. We have added this explanation in the calculation method.

The manuscript was modified as follows:

Page 19:

The spin-polarized density functional theory (DFT) calculations have been conducted on Vienna ab-initio simulation package (VASP)^{74,75} to study the alkaline hydrogen evolution reaction. The Projector augmented wave method⁷⁶ with a cutoff energy of 450 eV accompanied by Perdew-Burke-Ernzerhof functional⁷⁶ has been used in the DFT calculations. DFT + U method⁷⁷ with the effective U value of 3.0 eV⁷⁸ and DFT-D3 method⁷⁹ was used to correct the influence of 3d electrons of V atom and van der Waals interactions, respectively. Two layers of V_2O_5 (001) facet have been cleaved with a vacuum layer of 15 Å to build the V_2O_5 slab model and half of bottom layers have been fixed to simulate the bulk phase. Four of Mg-atom, Ca-atom and Sr-atom with 16 of H_2O molecules have been inserted in the interlamination of V_2O_5 , respectively to build the MgVO, CaVO and SrVO models. Besides, 16 of H_2O molecules have been inserted in the interlamination of V_2O_5 to build the $V_2O_5 \cdot nH_2O$ model. All models have been fully relaxed with the energy convergence criterion of 10^{-5} eV and the force convergence criterion of 0.02 eV/Å, respectively. The free energy of H^+ is obtained by $H^+ + e^- \rightarrow 0.5 H_2$ according to the National Institute of Standards and Technology chemistry webbook and the free energy of TMB- H^+ has been calculated by regarding TMB- H^+ as a radical. The $(1 \times 2 \times 1)$ K-points have been used as K-point mesh. The adsorption energy (E_{ads}) has been calculated using formula 1,

$$E_{ads} = E_{total} - E_{substrate} - E_{adsorbate} \quad (1)$$

The E_{total} , $E_{substrate}$ and $E_{adsorbate}$ represent the energy of adsorption structure, substrate and adsorbate, respectively. The free energies have been calculated using the following formula 2,

$$G = E_{DFT} + ZPE - TS \quad (2)$$

The G , E_{DFT} , ZPE and TS represent the free energy, energy from DFT calculations, zero point energy and entropic contributions, respectively.

Reviewer #3 (Remarks to the Author):

This manuscript reported a series of $M_xV_2O_5 \cdot nH_2O$ peroxidase mimics with vanadium as the redox site and redox-inert alkaline-earth metal ion (M^{2+}) as the Lewis acid, which have more significant peroxidase-mimicking activity compared with other typical peroxidase mimics according to the maximal reaction velocity (V_{max}) the turnover number (TON). The mechanism has been analyzed carefully by experiments and theoretical calculations. On this basis, $M_xV_2O_5 \cdot nH_2O$ was used for antibacterial treatment and achieved remarkable effect. But some shortcomings also should be further corrected in the manuscript. I prefer to accept it after major revision. Some comments are listed as follows:

1. The quality of Fig 1C and D should be improved. Spherical aberration electron microscope is recommended.

Response: According to your suggestion, we have resupplied the high-resolution transmission electron microscope images of $M_xV_2O_5 \cdot nH_2O$ and replaced Fig 1C and D with them. These new images can well demonstrate the structural information of the synthesized $M_xV_2O_5 \cdot nH_2O$. It should be noted that the abundant vacancy defects in these materials may lead to disruption of the crystal lattices, which made the surfaces of the materials look more irregular. In addition, we also added AFM images (Supplementary Fig. 4) and TEM images (Supplementary Fig. 2) with large field of view to better show the structure of these materials.

The manuscript was modified as follows:

Page 5:

Fig. 1 (C) HRTEM images (the inserts are SAED images) with clear lattice fringes and lattice disorder (D) of MgVO (a), CaVO (b) and SrVO (c).

The data were added in supporting information:

Supplementary Fig. 2 The TEM images with a large field of view of MgVO (A), CaVO (B) and SrVO (C).

Supplementary Fig. 4. AFM images (left) with height profile (right) of MgVO (A), CaVO (B) and SrVO (C). (a), (b) are two representative materials, respectively.

2. The biological safety of materials is very important, and it is suggested that the authors examine the safety of materials on normal cells, such as skin cells, vascular endothelial cells, fibroblasts, and so on. These experiments must be rigorous and substantial.

Response: Thank you very much for your reminding. We have added the cell viability tests using human dermal microvascular endothelial cells (HMEC-1) and fibroblasts cells (L929) as models in the revised manuscript. As shown in Supplementary Fig. 30, $M_xV_2O_5 \cdot nH_2O$ show good biological safety. In addition, due to the stable gel structure coated on the $M_xV_2O_5 \cdot nH_2O$, $M_xV_2O_5 \cdot nH_2O$ usually play an antibacterial role on the surface of the wound rather than entering the body. The large size of $M_xV_2O_5 \cdot nH_2O$ can also inhibit its uptake by cells.

The manuscript was modified as follows:

Page 15:

In addition to antibacterial properties, the banded structure of $M_xV_2O_5 \cdot nH_2O$ are widely believed to contribute to the formation of stable composites. Meanwhile, the large size reduces cytotoxicity due to the difficulty in endocytosis (Supplementary Fig. 30).

The data were added in supporting information:

Supplementary Fig. 30 Assay of the viability of HMEC-1 (A) and L929 (B) cells cultured with varying $M_xV_2O_5 \cdot nH_2O$ concentrations.

3. I can not identify $M_xV_2O_5 \cdot nH_2O$ nanomaterials in SEM images in Fig. S18. And the quality of SEM images of hydrogels were poor.

Response:

According to your suggestion, we re-tested the composite hydrogel by SEM. Since the GelTA is the main component of the composite hydrogel, $M_xV_2O_5 \cdot nH_2O$ are only low content additives, it is difficult to directly observe the $M_xV_2O_5 \cdot nH_2O$ in the composite hydrogel. In the revised support information, we replaced the previous SEM images with the SEM images containing EDX elemental quantification. The new data can show the elemental composition of the $M_xV_2O_5 \cdot nH_2O$ functionalized composite hydrogel, as well as more typical porous structures.

The manuscript was modified as follows:

Page 15:

SEM images showed that freeze-dried GelTA has abundant porous structure with high interconnectivity. $M_xV_2O_5 \cdot nH_2O$ functionalized GelTA (Mg-Gel-TA/ Ca-Gel-TA/ Sr-Gel-TA) also displayed similar porous structure, which were beneficial for exchange of substance (Supplementary Fig. 32). EDX elemental quantification shows the existence of $M_xV_2O_5 \cdot nH_2O$ in the composite hydrogel.

The supporting information was modified as follows:

Supplementary Fig. 32 The SEM images of GelTA (A) and SEM images with EDX elemental quantification (insets) of Mg-GelTA (B), Ca-GelTA (C), Sr-GelTA (D).

4. The bacteria in the wound sites of different treatment groups need to be further characterized.

Response: In order to verify the relationship between wound recovery and the number of bacteria in different treatment groups in a certain time, the standard spread plate method was used to culture the bacteria from the wound sites and then the bacteria are counted. The statistical results added in Supplementary Fig. 35 showed the number of bacteria from the wound sites of mice treated with $M_xV_2O_5 \cdot nH_2O$ functionalized GelTA significantly decreased, indicating that $M_xV_2O_5 \cdot nH_2O$ functionalized GelTA possess obvious antibacterial activity.

The manuscript was modified as follows:

Page 17:

In addition, the bacteria from wound tissues of mice in each group were cultured on LB agar plates, and the results showed that the number of bacteria significantly decreased at the wound sites in the treatment groups (Fig. 6E, Supplementary Fig. 35).

The data were added in supporting information:

Supplementary Fig. 35 Bacteria number in different groups after treatment at day 5. Error bars represent \pm SD (n = 3).

5. What is the mechanism of material sterilization? Can hydroxyl radicals cross bacterial cell walls? Bacteria are more resistant to free radicals than mammalian cells due to the protection of cell wall. If the free radicals produced by the material are able to kill the bacteria, could the damage to the surrounding normal skin cells be greater? How to avoid damage to the normal cells of the wound? In addition, free radicals can cause wound inflammation. How to avoid the wound inflammation caused by free radicals?

Response: It has been reported that the oxidative attack from the exterior to the interior of the bacteria by hydroxyl radicals is the main antibacterial mechanism of related nanomaterials (Biomaterials, 2010, 31, 7526). Specifically, the hydroxyl radical first destroys the cell wall of bacteria, forming pits and holes in them, and eventually leading to the leakage of the intracellular substances and bacteria inactivation (Environ. Sci. Technol., 2012, 46, 4042).

About the damage to the normal cells:

As with most ROS based antibacterial materials, ROS produced by the $M_xV_2O_5 \cdot nH_2O$ will inevitably damage the cells close to the hydrogel on the wound surface, but the death and regeneration of these cells is a common phenomenon in the wound healing process. In the group treated by $M_xV_2O_5 \cdot nH_2O$ functionalized GelTA, the wound healing was obvious (Fig. 6B), the epidermal layer (Fig. 6F) and collagen fibers (Fig. 6G) were intact, indicating that $M_xV_2O_5 \cdot nH_2O$ in the hydrogel had no obvious killing effect on tissue cells.

About the wound inflammation:

The composite hydrogel exists as a wound dressing, essentially creating a barrier between the environment and the wound. $M_xV_2O_5 \cdot nH_2O$ act as antibacterial additives to kill bacteria that try to enter the wound surface. Due to the stability of the hydrogel and the large size, $M_xV_2O_5 \cdot nH_2O$ does not tend to enter tissues and cause systemic oxidative stress, which can be confirmed by Fig. 6F. There is no significant

inflammatory reaction in the healed tissues of mice in the treatment groups. The above content has been discussed in the article.

Page 14, 15:

The enzyme mimics are widely used in antibacterial researches, because the reactive oxygen species (ROS) catalyzed by enzyme mimics can oxidize key components of bacteria, such as cell membranes/walls or intracellular compartments⁷⁰⁻⁷².

Page 17:

Further, Hematoxylin and eosin (H&E) staining was analyzed for the wound tissue of different groups of mice after seven days of treatment (Fig. 6F). The results show that the epidermal layers at the wound sites are more intact in the treatment groups, and only a few inflammatory cells are distributed. In the control groups, the epidermal layers at the wound sites are fragmented and many inflammatory cells gather in the wound areas. Masson staining shows that the density of collagen fibers at the wound site are higher in the treatment groups than in the control groups (Fig. 6G).

Reviewers' Comments:

Reviewer #1:

Remarks to the Author:

The authors have addressed all the comments, suggestions and revised the manuscript thoroughly. Hence, the paper can be accepted for publication.

Reviewer #2:

Remarks to the Author:

The authors have thoroughly addressed all my queries. The mechanistic aspects are explained in depth and comprehensively. I recommend publishing the paper in its current form.

Reviewer #3:

Remarks to the Author:

The authors have carefully revised the manuscript according to the reviewers' comments. The quality and novelty of this manuscript are greatly improved, and I fully agree to publish it.